# Proteome-Wide Identification of RNA-dependent proteins and an emerging role for RNAs in *Plasmodium falciparum* protein complexes

Thomas Hollin [1], Steven Abel[1], Charles Banks[2], Borislav Hristov[3], Jacques Prudhomme [1], Kianna Hales[3], Laurence Florens [2], William Stafford Noble [3,4] & Karine G. Le Roch [1] ✉

Ribonucleoprotein complexes are composed of RNA, RNA-dependent proteins (RDPs) and RNA-binding proteins (RBPs), and play fundamental roles in RNA regulation. However, in the human malaria parasite, *Plasmodium falciparum*, identification and characterization of these proteins are particularly limited. In this study, we use an unbiased proteome-wide approach, called R-DeeP, a method based on sucrose density gradient ultracentrifugation, to identify RDPs. Quantitative analysis by mass spectrometry identifies 898 RDPs, including 545 proteins not yet associated with RNA. Results are further validated using a combination of computational and molecular approaches. Overall, this method provides the first snapshot of the *Plasmodium* protein-protein interaction network in the presence and absence of RNA. R-DeeP also helps to reconstruct *Plasmodium* multiprotein complexes based on co-segregation and deciphers their RNA-dependence. One RDP candidate, PF3D7_0823200, is functionally characterized and validated as a true RBP. Using enhanced crosslinking and immunoprecipitation followed by high-throughput sequencing (eCLIP-seq), we demonstrate that this protein interacts with various *Plasmodium* non-coding transcripts, including the *var* genes and *ap2* transcription factors.

Ribonucleoprotein (RNP) complexes are critical post-transcriptional regulators of gene expression covering all aspects of RNA activity in eukaryotes such as export, splicing, stability, translation and degradation[1-3]. These RNPs are assemblies of RNA molecules and proteins including RNA-binding proteins (RBPs), and their compositions are highly dynamic to allow adaptation to cellular needs and environmental conditions[4]. RNPs contain RNAs ranging from messenger RNAs (mRNAs) to non-coding RNAs (ncRNAs) such as long ncRNAs (lncRNAs), rRNAs and tRNAs[2,5-7]. These RNAs can interact with RBPs and act in (post-)transcriptional and (post-)translational regulation, modulate the structures and stability of RNP complexes, or serve as protein decoys[2]. The advent of lncRNA research highlighted the involvement of these transcripts in post-transcriptional control. Recently, lncRNAs have been shown to be preponderant mediators in RNP complexes[2], and elucidating the composition and function of such complexes is a current challenge in RNA biology.

[1]Department of Molecular, Cell and Systems Biology, University of California Riverside, Riverside, CA, USA. [2]Stowers Institute for Medical Research, Kansas City, MO, USA. [3]Department of Genome Sciences, University of Washington, Seattle, WA, USA. [4]Paul G. Allen School of Computer Science and Engineering, University of Washington, Seattle, WA, USA. ✉e-mail: karine.leroch@ucr.edu

On the other hand, RBPs are also essential components of RNP complexes. They bind RNA through one or multiple RNA-binding domains (RBDs) such as the RNA recognition motif (RRM), K homology (KH), zinc finger, Pumilio (Puf), and DEAD box helicase domains[8]. In humans, 2000–3000 RBPs have been identified through several RNA interactome studies[9,10], while around 1000 are annotated in various model organisms such as *Mus musculus* and *Saccharomyces cerevisiae*[10]. In *Plasmodium falciparum*, the deadliest human malaria parasite[11], RBPs also regulate a wide range of essential processes[12–17], but our knowledge gaps in identifying and characterizing their role in the parasite represent a critical roadblock in the fight against malaria. To date, there are only two in silico approaches and one mRNA interactome capture dataset available in *P. falciparum*. The first bioinformatics analysis was published in 2015 and reported 189 putative RBPs[18]. These proteins belonged to 13 RBP families, including some of the most prominent, such as RRM, KH and zinc finger domain. The RBP repertoire of *P. falciparum* was then expanded with a hidden Markov model search using 793 RNA-related domains[19]. A total of 988 putative RBPs were identified and corresponded to 18.1% of the parasite proteome. This study also included an experimental capture of mRNA-binding proteins (mRBPs) using oligo d(T) beads followed by mass spectrometry identification. The authors captured 199 candidate mRBPs, with an enrichment of RRM, DEAD, LSm and Alba domains.

These two proteome-wide studies have provided the most complete core set of RBPs in *P. falciparum*. However, the discovery of unconventional RBPs, generally lacking canonical RBDs, highlight the great challenges to comprehensively identify RBPs within eukaryotic proteomes[10]. Developing unbiased and complementary RNA interactome approaches are therefore necessary to elucidate the RBP repertoire and composition of RNP complexes to facilitate our understanding of their biological functions. Recently, a quantitative proteome-wide screen based on density gradient ultracentrifugation (R-DeeP) was developed to identify RBPs as well as RNA-dependent proteins (RDPs) within RNP complexes. RDPs are proteins that do not bind directly to RNA but interact with other RDPs or RBPs and are crucial components of RNP complexes. Applying this approach, the authors identified 1784 RDPs, 537 of which had never been associated with RNA in human HeLa cells[20]. More recently, a total of 1189 candidates, including 170 unknown RDPs, were detected in A549 lung adenocarcinoma cells[21]. Here, we applied the R-DeeP method to *P. falciparum* and identified 898 RDP candidates, including uncharacterized proteins. Furthermore, we demonstrated that this approach can be used to interpret *Plasmodium* complexes and protein clusters, and identify RNP complexes. Finally, we experimentally characterized the protein PF3D7_0823200 using complementary approaches, including high-throughput sequencing of RNA isolated by crosslinking immunoprecipitation (eCLIP-seq), and proposed this protein as part of a potential RNA stabilization and/or splicing factor complex.

## Results

### Identification of RNA-dependent proteins using R-DeeP

The R-DeeP method is based on the separation of proteins on a sucrose density gradient in the presence (= Control) or absence (= RNase) of RNA (Fig. 1a). The separation of proteins or multiprotein complexes depends on their respective molecular weights (MW), with larger proteins or complexes found in higher density fractions. For RNA-dependent proteins (RDPs), the RNase treatment may impact their interactome and thus result in a shift towards fractions of lower sucrose concentration. In this study, we extracted mid-late trophozoites by saponin lysis and prepared soluble protein extracts in RNase-free conditions (Fig. 1a). After DNase I treatment (for Control samples) or DNase I and RNase A/H/I treatment (for RNase samples) (Supplementary Fig. 1a), 2–2.5 mg of proteins were loaded onto a 5–50% sucrose density gradient. After ultracentrifugation, 25 fractions (numbered from the top of the gradient

to the bottom) were collected for each condition and analyzed by mass spectrometry or western blot.

First, we generated Control and RNase samples in duplicate and quantified the protein abundance using MudPIT mass spectrometry (Fig. 1b). After filtering and normalization, we generated a final list of 3671 proteins reproducibly detected in each sample, representing > 66% of the proteins encoded in *Plasmodium* (5545 proteins in the database used) (Supplementary Data 1). For each fraction the amount of each protein was assessed in each replicate and a Pearson correlation coefficient was then computed. As we obtained a value of 0.702, we concluded that results obtained were reproducible and we therefore combined our experiments for further downstream analysis. To identify RDPs, we calculated the cumulative distribution function (CDF) for each protein's abundance across the 25 fractions and used a Wilcoxon rank-sum test to detect proteins that exhibit a statistically significant shift (Figs. 1c and 1d). A total of 898 unique proteins were identified as significantly left-shifted, suggesting that their interactions are RNA-dependent (Fig. 1e and Supplementary Data 1). Additionally, 49 proteins were detected as right-shifted, including 14 *rifin* genes, but no particular pathway seemed to be associated with them (Supplementary Data 1). These proteins may have interacted with newly accessible partners in the absence of RNA. For the 898 left-shifted proteins, Gene Ontology (GO) enrichment analysis showed a strong enrichment for diverse RNA pathways such as mRNA processing, mRNA metabolic process, RNA splicing, and ribosome biogenesis (Fig. 1f), confirming the robustness of our R-DeeP experiment.

We next generated a list of proteins already defined as RNA-associated proteins, including RNA-binding proteins (RBPs), using PlasmoDB's GO resource[22], two in silico datasets[18,19], and an mRNA interactome capture experiment[19]. A collection of 1319 unique RNA-associated proteins were obtained (Supplementary Data 2) and compared to our R-DeeP list. A total of 39% (353/898) of our shifted proteins were previously associated with RNA, which included 23 ribosomal proteins, 16 RNA helicases, and 8 LSm proteins, as well as 19 poorly characterized *Plasmodium* proteins and 19 putative RBPs (Fig. 2a). Several proteins associated with stress granules and P-bodies, cytoplasmic condensates of RNA and proteins, were detected as shifted, including CITH, CAF1, and CAF40 (Supplementary Data 1). Additionally, 26 proteins were detected in all experimental and computational datasets and correspond to well-known RBPs such as PRP22, Alba 2 and 4, CUGBP Elav-like family member 1 and 2, and polyadenylate-binding protein 3. Further comparisons showed that among the 353 RNA-associated proteins shared between R-DeeP and other datasets, 244 proteins (69%) were previously identified in at least two other datasets (Fig. 2b), confirming the robustness of our R-DeeP experiment.

Among the RNA-associated complexes depleted in our experiment, we noticed an enrichment of 142 ribosomal proteins, 30 (eukaryotic) translation initiation factors, 20 pre-mRNA splicing factors, 14 elongation factors and 9 exosome components (Fig. 2a). In the human R-DeeP experiment[20], the majority of the translation initiation factors were found as not shifted, confirming that proteins can be involved in RNA processes and not being RNA-dependent for the formation and/or stability of their respective complexes. For the human ribosome subunits and exosome, their proteins were identified as shifted, highlighting some discrepancies with our data. As they are both well-known RNP complexes, we can hypothesize that in our experimental conditions, the RNA may be protected from RNase activity and only ribosomal proteins located on the surface of the complex, such as RPL29 (PF3D7_1460300) and RPS12 (PF3D7_0307100), could have been detached from the rest during the RNase treatment (Supplementary Fig. 1b). Thus, the detection would be limited for large RNP complexes having RNA embedded inside their structure and not accessible to RNases.

As the R-DeeP screen does not only identify classical RBPs but also unconventional RBPs and RDPs, we suggest that some of 545 proteins unique to the R-DeeP dataset could be RDPs or unknown RBPs. Among

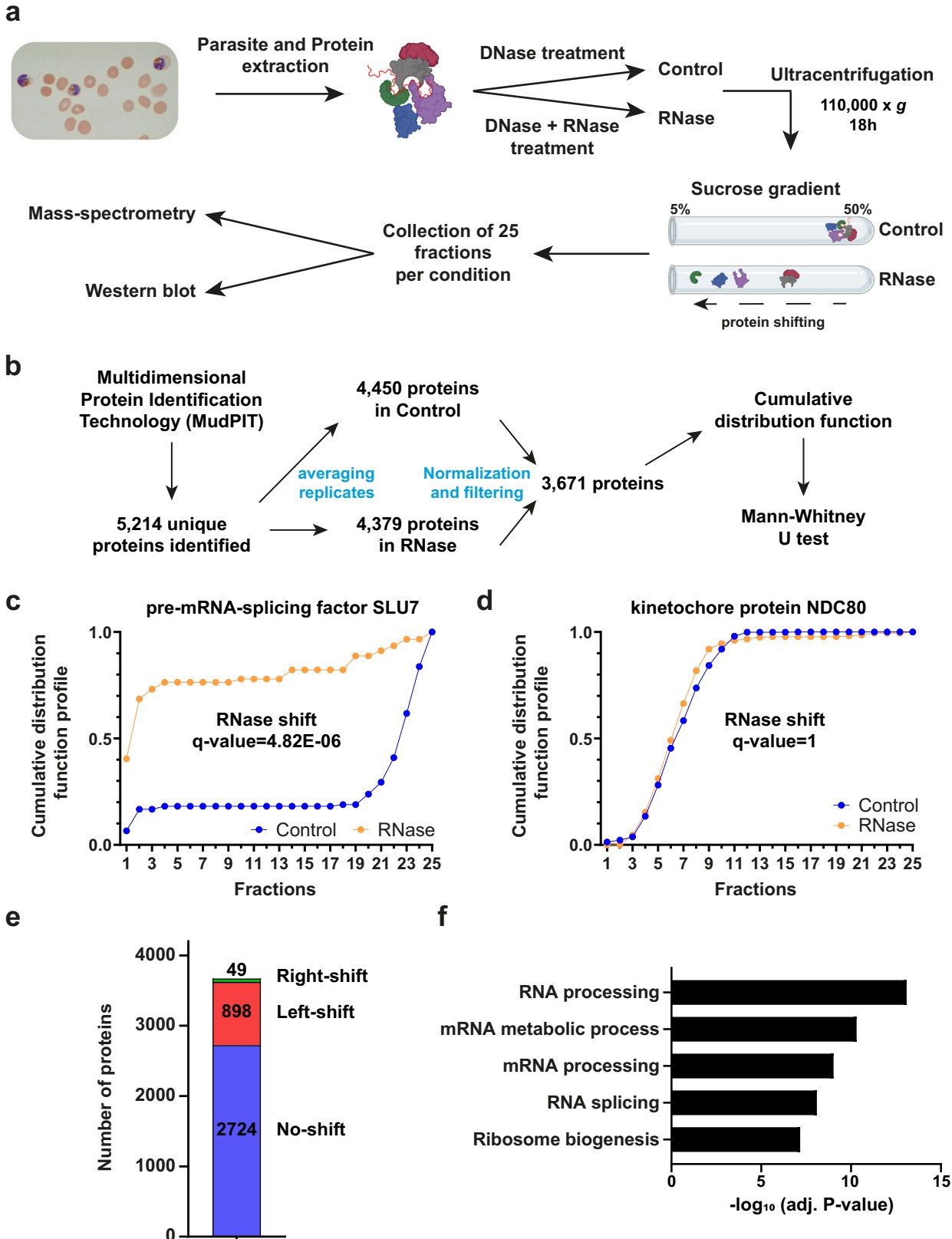

these proteins, we identified 181 uncharacterized proteins of which 138 are annotated as *Plasmodium* specific. This cluster also contains seven heptatricopeptide repeat (HPR) proteins which are related to pentatricopeptide repeat (PPR) proteins, a well-known RBP family in land plants. In *P. berghei*, a malaria rodent model, PbHPR1 binds in vitro to mitochondrial RNAs, suggesting that these proteins are bona fide

RBPs[16]. Interestingly, 14 AP2 transcription factors were also significantly shifted, indicating that the stability of their respective complex may depend on transcriptional activity. Considering that these 545 potential RDPs may interact with other RDPs or RBPs, we analyzed a recent publication that claimed to identify over 20,000 putative protein interactions in *Plasmodium*, comprising 1259 unique proteins,

**Fig. 1 | R-DeeP approach to identify RNA-dependent proteins in *P. falciparum*.**
**a** Schematic overview of the R-DeeP method. NF54 parasite protein lysates were
treated with DNase (Control) or DNase + RNases (RNase) and loaded on a sucrose
gradient. After ultracentrifugation, 25 fractions were collected and further pro-
cessed by mass spectrometry and western blot analysis. Created with BioR-
ender.com. **b** Bioinformatics workflow for the mass spectrometry data analysis.
After multiple filtering (see Methods), a final list of 3671 proteins was obtained and a

cumulative distribution function (CDF) was calculated for each protein. CDF pro-
files of pre-mRNA-splicing factor SLU7 (**c**) and kinetochore protein NDC80 (**d**)
illustrate an RNase-shifted and non-RNase-shifted protein, respectively. **e** The
graph shows the number of left-shifted, right-shifted and non-shifted proteins
detected in this R-DeeP. **f** GO enrichment analysis of the 898 left-shifted proteins.
The significance of Biological Process terms is shown by $-\log_{10}$ (adjusted *P*-value)
(Fisher's exact test with Bonferroni adjustment).

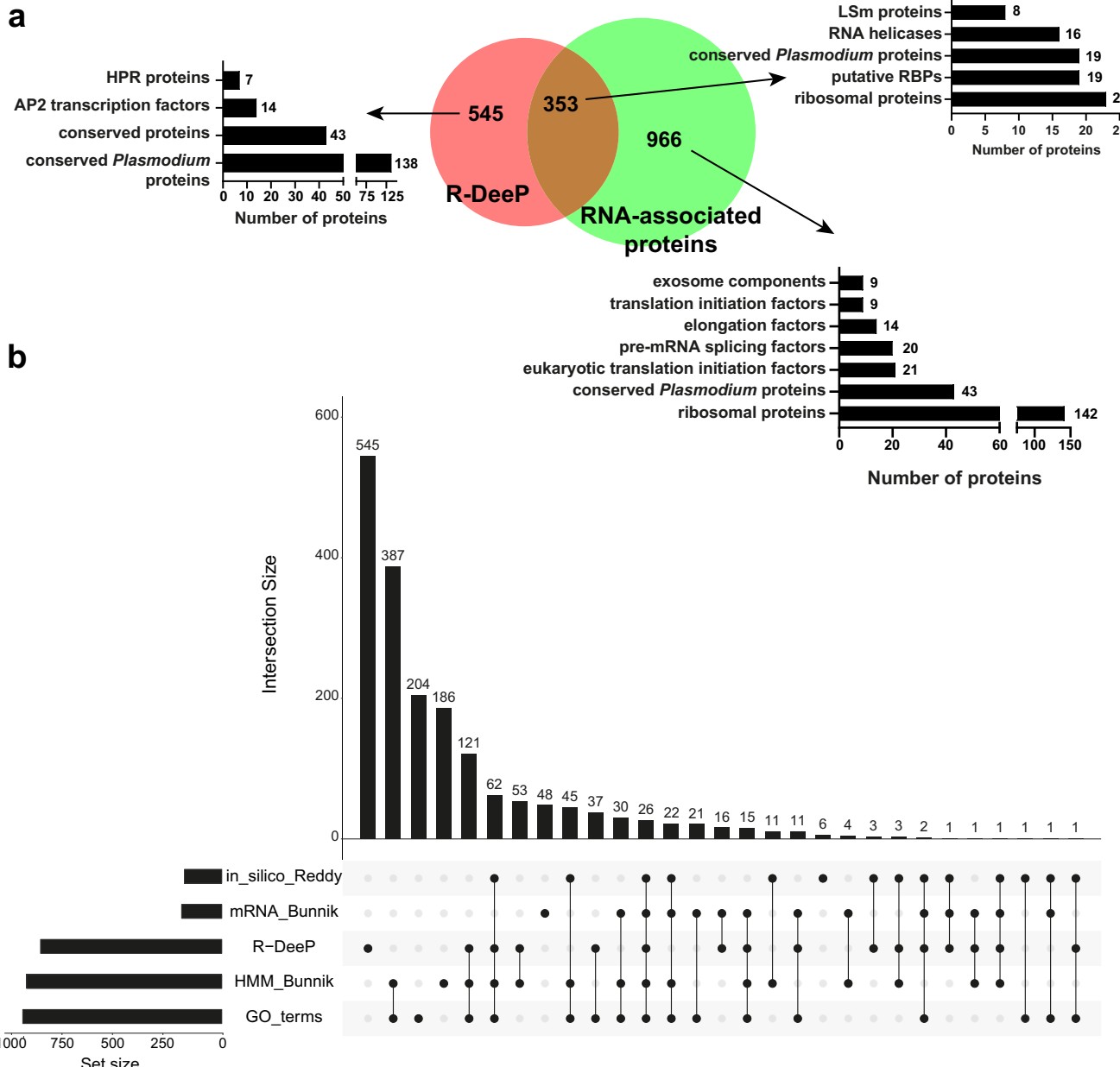

**Fig. 2 | Comparison of the significant left-shifted proteins. a** Venn diagram
reporting the number of proteins overlapping between the RDPs identified in this
R-DeeP and a custom list of known RNA-associated proteins (see Methods). For
each cluster, some of the most represented protein families/groups are indicated.

**b** UpSet plot summarizing the number of unique and shared RNA-dependent
proteins between different datasets (in_silico_Reddy[18]; mRNA_Bunnik[19]; R-DeeP
from this study; HMM_Bunnik[19]; GO_terms from PlasmoDB[22]).

using quantitative mass spectrometry and machine learning[23]. Among
the 109 proteins clustered into putative protein complexes, 71 proteins
were associated with either known RBPs or proteins significantly
shifted in our R-DeeP experiment (Supplementary Fig. 1c and Data 2).
To further investigate the likelihood of these 545 candidates to be
associated with RNA, we took advantage of the Enzyme Commission

(EC) numbers provided by OrthoMCL, a database grouping ortholo-
gous protein sequences[24]. A total of 109 candidates were linked to an
EC number and 101 of them clustered at least with one ortholog pre-
viously identified as RBPs in various model organisms[25] (Supplemen-
tary Fig. 1c and Data 2). Altogether, only 147 of the 545 proteins interact
or share similarities with other RBPs, indicating that the majority of

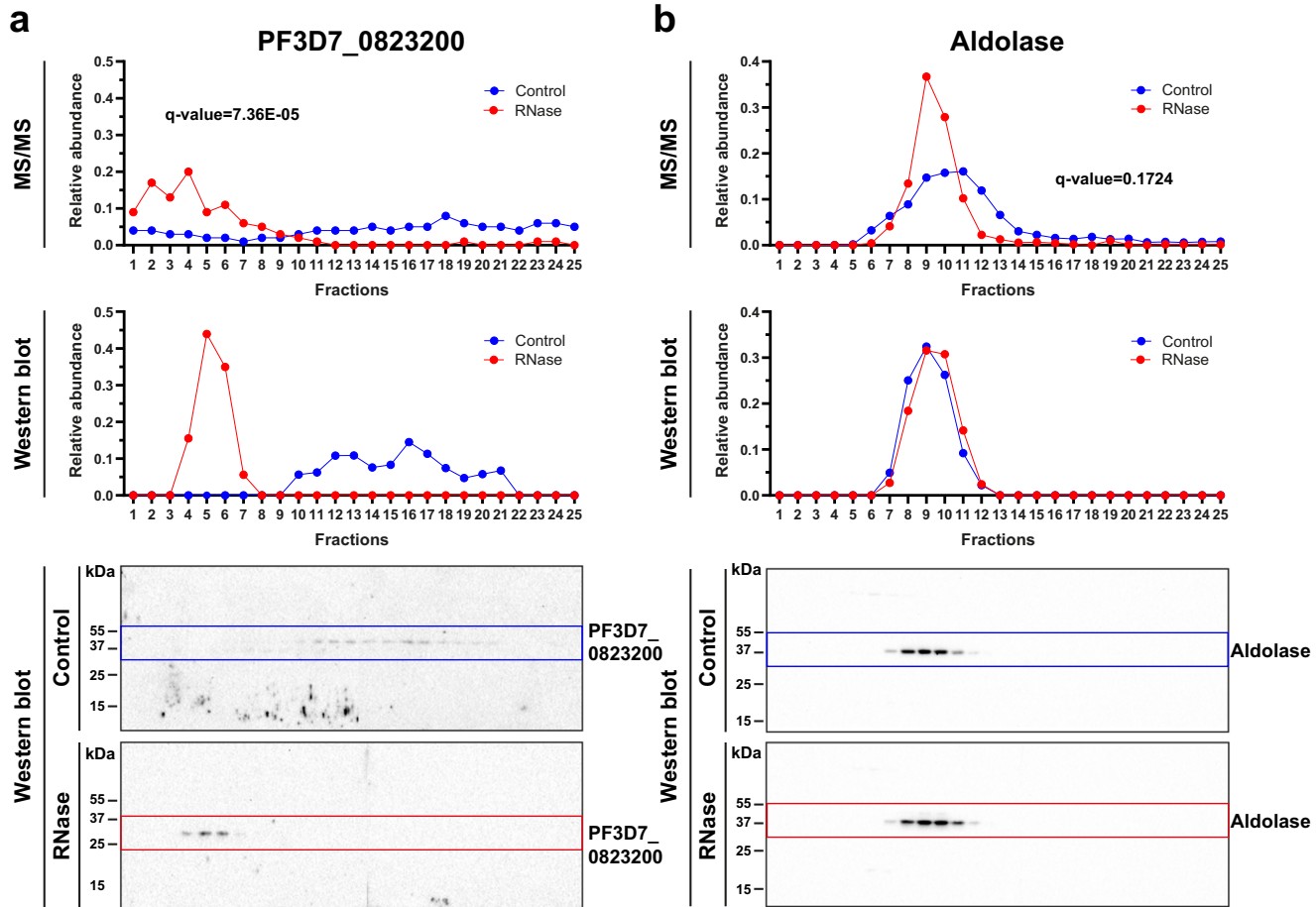

**Fig. 3 | Validation of the R-DeeP protocol by western blot analysis.** The mass spectrometry (MS/MS) data (top panel) were compared to the quantitative analyzes of immunoblots (center panel) obtained with anti-PF3D7_0823200 (**a**) and anti-Aldolase (**b**) (bottom panel).

these uncharacterized proteins may be potential RDPs or unconventional RBPs requiring further investigations.

## Validation of the R-DeeP screening

To further validate our R-DeeP results, we analyzed the profiles of some proteins using an independent R-DeeP replicate by western blot analysis. As the availability of commercial antibodies is limited in *P. falciparum*, we produced eight different custom rabbit polyclonal antibodies. We selected peptides targeting six significantly shifted proteins, including two unknown *Plasmodium* proteins (PF3D7_0528600 and PF3D7_1354900), two putative RBPs (PF3D7_1360100 and PF3D7_0823200) and two characterized RBPs (Musashi (PF3D7_0916700) and Alba 4 (PF3D7_1347500)) (Supplementary Data 3). Proteasome subunit alpha type-7 (PSA 7, PF3D7_1353900) and cytochrome c oxidase subunit 6 A (COX6A, PF3D7_1465000) were chosen as negative controls, as well as fructose-bisphosphate aldolase (PF3D7_1444800), for which a commercial antibody is available. The reactivity of these antibodies was tested on total parasite protein lysates, and only PF3D7_0823200, PF3D7_1347500 and PF3D7_1353900 successfully showed specific recognition (Supplementary Fig. 2a). Immunoblots of the 25 R-DeeP fractions with and without RNase treatment were carried out using these different antibodies as well as the anti-Aldolase. Protein signals were normalized for both conditions to obtain relative abundance and then matched to the mass spectrometry distribution profiles (Fig. 3). We validated the shifting of PF3D7_0823200, while the aldolase immunoblots reflected our mass spectrometry results, thus supporting the conclusion that this is an RNA-independent protein. Similarly,

immunoblots confirmed Alba 4 and PSA 7 as RNA-dependent and RNA-independent proteins, respectively (Supplementary Figs. 2b, c). Interestingly, a difference of migration was observed for PF3D7_0823200 between Control (~ 30 kDa, in agreement with its calculated MW of 32,310 Da) and RNase (~ 45 kDa) conditions. This discrepancy might be attributable to post-translational modifications or dimerization only occurring when the protein is part of its RNP complex, but this requires further investigation.

## The fate of protein-protein interaction networks in the presence and absence of RNA

With the R-Deep methodology, the position of each protein in the sucrose gradient is determined by its respective MW, structure, and interactome. Based on this information, we determined the network status of each protein in the Control and RNase samples. To do this, we calculated the halfway value indicating at which fraction 50% of the total amount of each protein was detected (CDF = 0.5). Using a previous R-DeeP calibration generated with human reference proteins (RNase A, BSA, Aldolase, Catalase and Ferritin)[20] and these halfway values, we were able to determine an apparent MW for all proteins. Then, the ratio between apparent and theoretical MW was used to classify the proteins according to their molecular state. Proteins appearing to be smaller, identical, or larger than their theoretical MW are indicated as 'smaller', 'monomeric' and 'larger', respectively (Supplementary Data 4).

For the Control condition, we identified 2790 proteins (76%) with a higher apparent than theoretical MW (ratio > 2) suggesting that they were in complex (Fig. 4a). Only 283 (7.7%) and 595 (16.2%) proteins were detected as smaller and monomeric, respectively. By contrast, in

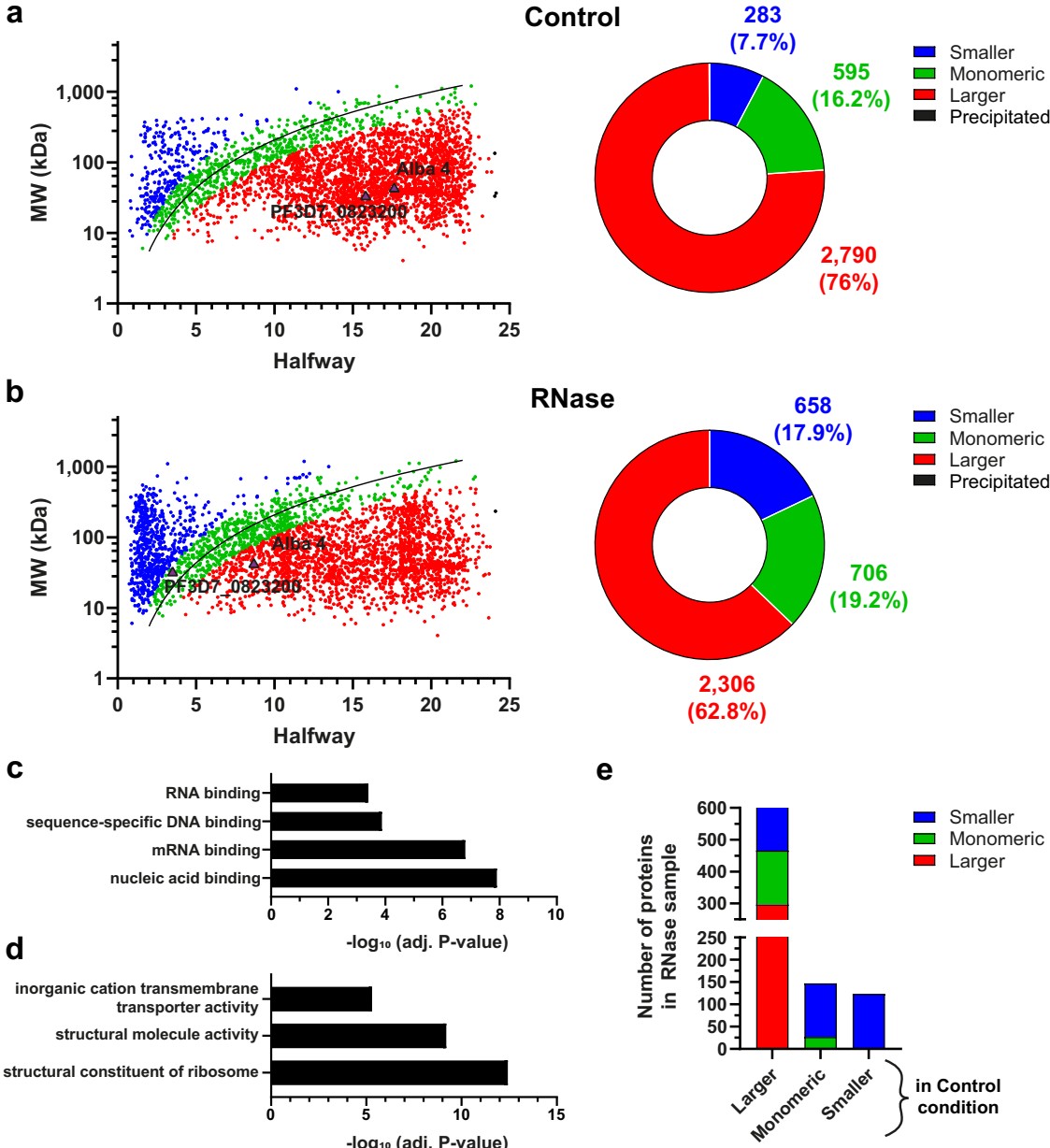

**Fig. 4 | Interaction networks prior and after RNase treatment.** Plots showing the halfway value and the theoretical molecular weight (MW) for each protein in Control (**a**) and RNase condition (**b**) (Left panel). Pie charts indicating the number and percentage of proteins considered smaller (blue), monomeric (green), larger (red) and precipitated (black) (Right panel). The reference extrapolation was used to calibrate the sucrose gradient and classify each protein into the different categories based on their MW ratio. The exact positions of Alba 4 and PF3D7_0823200 are indicated in control and RNase conditions. **c**, **d** GO enrichment analysis of proteins considered smaller and monomeric (**c**) and larger (**d**) after RNase treatment. The significance of Molecular function terms is represented as $-\log_{10}$ (adjusted *P*-value) (Fisher's exact test with Bonferroni adjustment). **e** Graph showing the displacement of the 898 significantly shifted proteins. These proteins were separated into three groups based on their classification in the control condition.

the RNase condition, we observed 658 (17.9%) smaller and 706 (19.2%) monomeric proteins (Fig. 4b). A clear shift can be noticed with 2306 (62.8%) larger proteins, instead of 76% in Control, a result similar to that obtained with HeLa cells (61%)[20]. GO enrichment analysis confirmed that non-complexed proteins (smaller and monomeric) are mainly involved in nucleic acid binding and mRNA/RNA binding (Fig. 4c). On the other hand, proteins categorized as still being in complex were mainly associated with ribosomes and cation transmembrane transporters such as V-type proton ATPases, ATP synthases and cytochromes (Fig. 4d).

Next, we focused on our list of 898 shifted proteins to evaluate their structural fate in the two different experimental conditions.

Interestingly, 329 out of 626 proteins considered as larger in the control condition were shifted to monomeric (169) and smaller (160) categories, indicating that the degradation of RNA resulted in loss of complex formation/stability (Fig. 4e). Among the shifted proteins remaining in complex in the RNase condition, we identified ribosomal and LSm proteins, suggesting that smaller modules of subunits may have formed from the full complexes. Moreover, 147 monomeric proteins were identified as monomeric in the Control, and 120 of them became smaller after RNase treatment (Fig. 4e). In total, 449/898 proteins (50%) were shifted to a lower category (larger > monomeric > smaller), confirming the deterioration of complex integrity for many of them. For PF3D7_0823200, the protein was shifted from larger to

monomeric with an apparent MW 17.98 and 0.6 times that of the theoretical in the Control and RNase condition, respectively (Fig. 4a, b). Although Alba 4 was still considered as co-sedimenting as part of a larger complex after RNase treatment, the MW ratio decreased significantly from 17.7 to 3.53, suggesting a partial shifting with the destabilization of the main large complex and the formation of smaller modules. Altogether, these results showed the benefit of R-DeeP profiling to decipher the fate of proteins in various biological conditions, including presence and absence of RNA.

### Investigation of multiprotein complexes in *P. falciparum*

Based on the previous observations, we hypothesized that the integrity of protein complexes is preserved in the control condition, except for those that are DNA-dependent. Thus, the interactions among members of a protein complex can be analyzed by co-segregation. Recently, functional associations between co-regulated human proteins were revealed using a machine learning algorithm, treeClust[26,27], which infers dissimilarities based on tree clustering. The authors showed that treeClust outperformed three correlation-based co-expression measures, including Pearson's correlation coefficient[26]. We compared these two measures using our R-DeeP data and found that the two scores performed very similarly (Supplementary Fig. 3). This discrepancy may be due to the fact that the human study was based on a larger sample pool (294 biological perturbations) in the context of predicting functional associations rather than co-complex membership. For the purposes of analyzing this dataset, we considered that the Pearson's correlation coefficient is sufficient.

Then, we first generated different random complexes ranging from 2 to 15 individual proteins and then assessed the mean Pearson's correlation coefficient of these proteins within the same complex. As expected, the correlation was low (−0.169 to 0.5, average 0.13) for these false protein complexes, especially for those with a large number of partners (Supplementary Data 5). Then we analyzed 47 different complexes identified in *P. falciparum* or conserved in eukaryotes, and considered as RNP complexes or not (Supplementary Fig. 4 and Supplementary Data 5). To visualize the protein–protein correlation, we generated a two-dimensional principal component analysis (PCA) map in which the distance between proteins indicates how similar their control profiles are. For five *Plasmodium* complexes, we observed that the proteins from a same complex tended to cluster together, indicating that our approach preserved protein complex information (Fig. 5a). Detailed analysis showed that the complexes involved in splicing such as U2, U6 (LSm), Prp19-CDC5 had a high correlation (> 0.8) as well as 20 S and 26 S proteasomes (Fig. 5b–d and Supplementary Data 5)[28,29]. The U6 RNP complex is nuclear and composed of a U6 small nuclear RNA (snRNA) and 7 LSm proteins (LSms 2-8). Conversely, LSm1 is not associated with U6 and is present in the cytoplasm, although it does interact with a low proportion of LSm proteins[30]. Co-segregation analysis of LSm1 confirmed the lower correlation between this protein and the U6 complex (0.71 vs 0.868) (Supplementary Data 5). Additional complexes also showed a good correlation (> 0.6), such as 60 S ribosomal subunits, mitochondrial complexes II and IV, RNA polymerase I and III, RNA exosome and mediator complex (Supplementary Fig. 4 and Supplementary Data 5). We also analyzed the spliceosomal B complex, which contains 46 proteins from the U2, U4-U6, U5 complexes as well as the Sm, LSm and CAP proteins. Although the mean correlation was low (0.254), we noticed 4 distinct clusters showing a higher correlation (Supplementary Fig. 4). The two main clusters mainly contained the U6 (LSm) and U4-U6 proteins, and the U2, U5, and Sm and CAP proteins, respectively.

Taking advantage of this method to reconstruct protein complexes using R-DeeP data, we analyzed the publication that claimed to identify over 20,000 putative protein interactions resulting in 593 clusters in *Plasmodium*[23]. Among a total of 593 clusters, with a mix of known and unknown complexes, we obtained a correlation coefficient for 442 of them containing at least two detected proteins (Supplementary Data 5). A subset of 113 clusters (25.6%) obtained a correlation > 0.6, suggesting that they are likely to be real complexes and require further investigation. This was the case, for example, with cluster 342 comprising two unknown proteins and displaying a correlation at 0.965, or cluster 410 containing 3 RNA polymerase I components and PF3D7_1454200, an unknown protein which could probably be associated with transcription based on its partners (Fig. 5e, f and Supplementary Fig. 5). Additional clusters showed lower correlations (Fig. 5b), but some are composed of two different subunits like cluster 25 with PA700 and eIF3 complexes, for which the separate correlations are high (Supplementary Fig. 5). Cluster 13 contains a total of 73 proteins, part of the 60 S and 40 S ribosomal subunits, and had a mean correlation at 0.666. Overall, these results confirm that our R-DeeP data can benefit the malaria community to elucidate known or hypothetical complexes, regardless of their RNA-dependence.

### RNA-dependence of *Plasmodium* ribonucleoprotein complexes

Reconstruction of different complexes in *P. falciparum* demonstrated that the proteins mostly remained structurally organized in the control condition, indicating that we can study the impact of RNase treatment at the complex level and not just the protein level. We calculated the significance for the different complexes by multiplying the independent shift *p*-values of each partner using Fisher's method. Thus, a *p*-value representative of RNA-dependence was assigned to the previous 47 complexes and 442 clusters (Supplementary Data 5). As expected, several complexes associated with splicing were significantly shifted such as U2 (*p*-value = 1.70E-11), U6 (LSm) (*p*-value = 3.90E-11), spliceosomal B complex (*p*-value = 1.60E-45) and Prp19-CDC5 (*p*-value = 0.0052) (Fig. 5c, Supplementary Fig. 4, and Supplementary Data 5). The RNA polymerase I and III complexes showed significant *p*-values at 0.0067 and 1.30E-12, respectively, confirming their dependence with RNA, as well as the mediator complex (*p*-value = 0.012), which is also associated with transcription[31]. For the 40 S ribosomal subunit, we noticed that the global correlation was low (0.441) and two distinct clusters of 23 and 8 proteins were formed (Supplementary Fig. 4). Although the complex showed a global *p*-value of 5.80E-07, confirming its RNA-dependence, 3 of the 4 shifted proteins identified were in the minor cluster. We can postulate that a small part of the 40 S subunit was detached during the experimental procedure, and that it was more sensitive to RNase treatment compared to the large particle. Pf60 S and RNA exosome were below the significance threshold (*p*-values at 0.0037 and 0.042, respectively), confirming that these complexes were disturbed by the by RNase treatment (Supplementary Data 5). For the 20 S proteasome, no significant shift was observed (*p*-value = 0.34), as well as mitochondrial complexes (*p*-values from 0.75 to 1), indicating, as expected, that their stabilities are not linked to RNA (Fig. 5d and Supplementary Data 5).

### PF3D7_0823200 is well conserved in *Plasmodium*

Given the R-DeeP mass spectrometry analysis, western blot validation, and the specificity of our custom antibody, we decided to further study the protein PF3D7_0823200. On the PlasmoDB database (v61)[22], this protein is annotated as a putative RBP since two RNA recognition motif (RRM) domains were identified by SMART and ScanProsite. In 2021, PF3D7_0823200 was described as ortholog of UIS2 protein (PBANKA_0506200), a critical RBP for gametocyte development and production of sporozoite in *P. berghei*[32]. However, although PF3D7_0823200 is the closest homolog of PBANKA_0506200 in *P. falciparum*, this first protein shares higher identity with PBANKA_0707400 (global identity 88% vs 7%), indicating that UIS2 is most likely not its ortholog. In fact, PF3D7_0823200 is well conserved in *Plasmodium* achieving >81% identity with proteins from various *Plasmodium* species, including *P. vivax*, *P. yoelii* and *P. chabaudi*

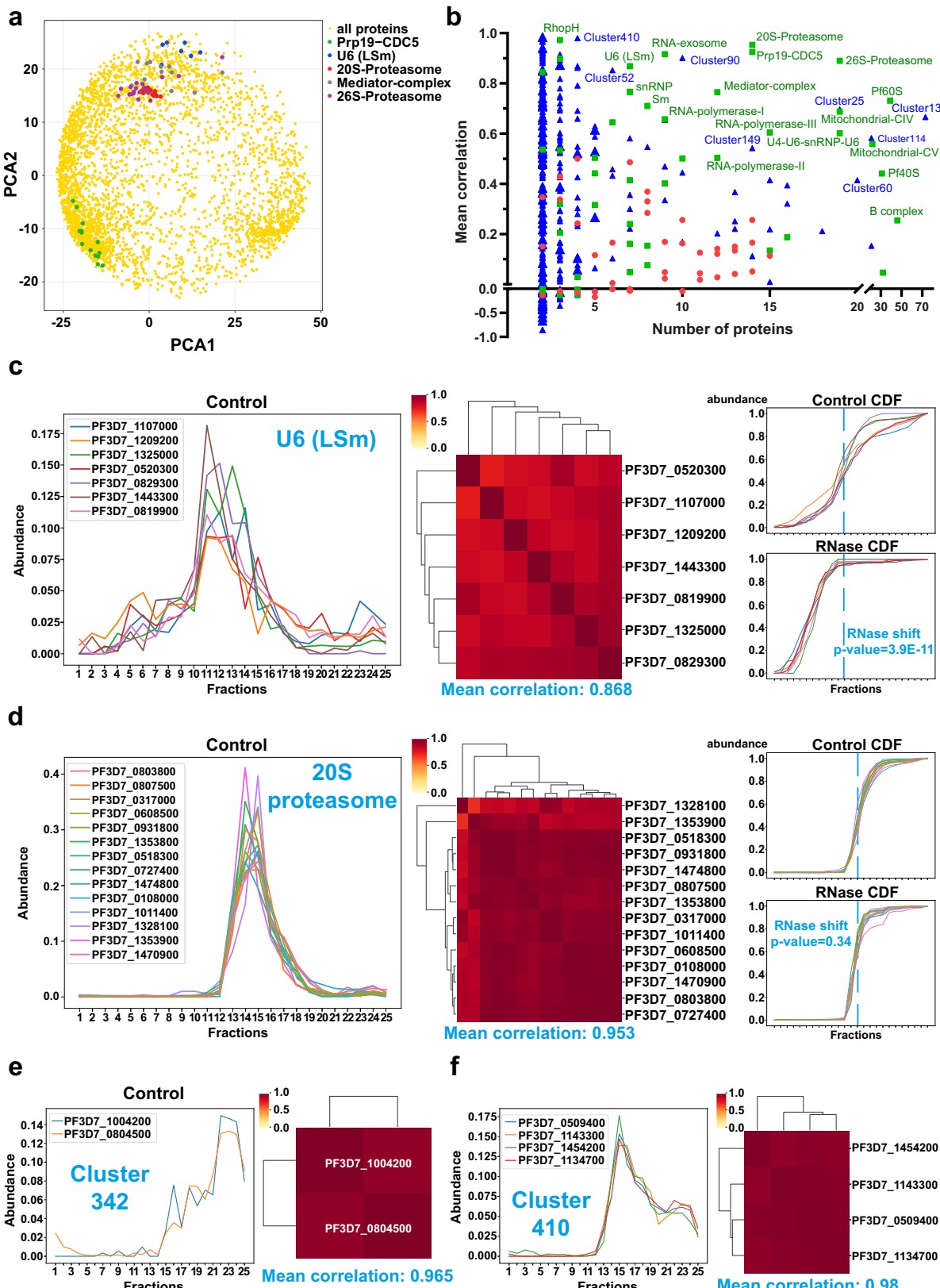

(Supplementary Data 6). This homology decreases considerably with other Apicomplexa but still matches uncharacterized RBPs in *Theileria* and *Neospora*. For *Cyclospora cayetanensis* and *Babesia microti*, two apicomplexan parasites, homology was detected (40% and 31% identity, respectively) with CUGBP Elav-like proteins, a family involved in pre-mRNA alternative splicing, mRNA stability, and translation[33].

**PF3D7_0823200 is a nucleo-cytoplasmic protein interacting with splicing and translational factors**

To further characterize PF3D7_0823200, we performed immuno-fluorescence assays (IFAs) to detect its localization in the parasite. We showed that PF3D7_0823200 was detected in all asexual stages of *P. falciparum*, and although the protein appeared to be enriched in

**Fig. 5 | Co-segregation of *Plasmodium* protein complexes and their RNA-dependence. a** 2-dimensional PCA representation of the protein quantification. Each protein detected is indicated in yellow and proteins belonging to five *Plasmodium* complexes with high correlation values are represented in different colors. These proteins cluster together, indicating that our approach preserves protein complex information. **b** Mean correlation of protein complexes in *P. falciparum*. Graph showing the mean correlation and number of proteins for all protein complexes investigated. These complexes are grouped in three different categories: random (red), *Plasmodium* (green) and from Hillier publication (blue). **c** Co-segregation and RNA-dependence of the U6 (LSm) complex. Graph showing the

mass-spectrometry profiles of all components of the U6 (LSm) complex under the control condition (left part). Using these different profiles, mean correlations were calculated and represented in the heatmap as well as an overall mean correlation for the U6 (LSm) complex (center part). Based on the cumulative distribution function (CDF) profiles and associated *p*-values for all components of the U6 (LSm) complex, an RNase shift *p*-value was calculated at the complex level (right part). RNase shift p-value was determined by multiplying associated *p*-values (one-sided t-test) and adjusted for multiple hypotheses using Fisher's method. **d** Co-segregation and RNA-dependence of the 20 S proteasome. **e, f** The control profiles, heatmaps and mean correlations are depicted for cluster 342 (**e**) and cluster 410 (**f**).

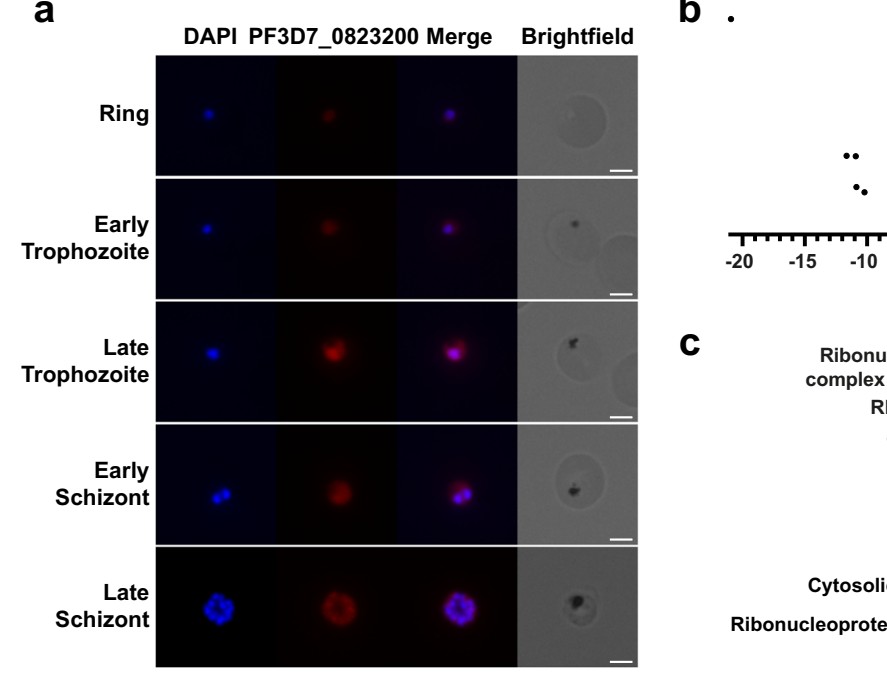

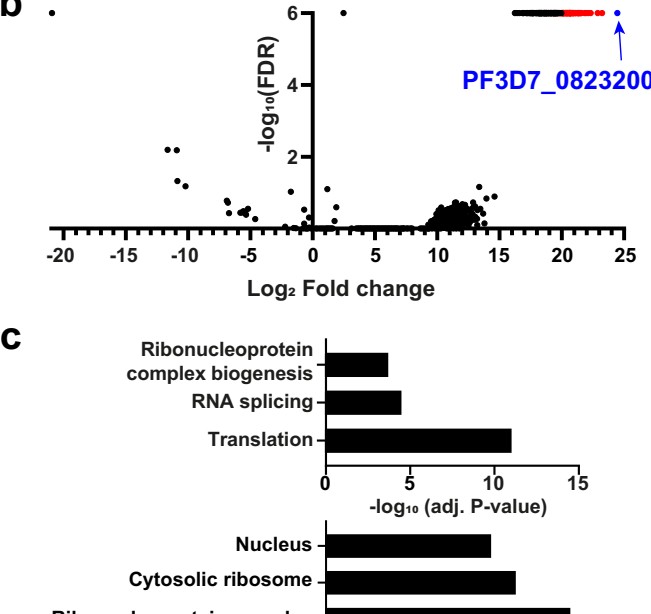

**Fig. 6 | Localization and interactome of PF3D7_0823200. a** Immunofluorescence assay of PF3D7_0823200 on ring, early trophozoite, late trophozoite, early schizont and late schizont. PF3D7_0823200 was labeled using its respective custom antibody and parasite nucleus was stained with DAPI. Merge shows both signals. Scale bar: 3 µm. *n* > 30 positive parasites from two independent experiments.
**b** Immunoprecipitation of PF3D7_0823200. Volcano significance plot highlights the

94 proteins significantly enriched (in red) in the three affinity purifications compared to three control purifications using anti-IgG. PF3D7_0823200 is highlighted in blue. **c** GO enrichment analysis of the significantly enriched proteins. The top 3 terms of Biological Process (top) and Cellular Component (bottom) are represented as −log₁₀ (adjusted *P*-value) (Fisher's exact test with Bonferroni adjustment).

the parasite nucleus, it was also present in the cytoplasm (Fig. 6a). This protein was previously identified in both cellular compartments by MS[34,35].

Based on the position of PF3D7_0823200 in the sucrose fractions, we established that the protein was part of a large complex (MW ratio = 17.98) in the control condition. Thus, to determine its potential partners, we performed immunoprecipitation followed by mass spectrometry (IP-MS) using the anti-PF3D7_0823200 antibody on soluble protein extracts of 3D7 parasites. Proteins were filtered with QPROT-calculated Log2 fold change >3, FDR < 0.01 and mean dNSAF ≥ 0.0005 compared to values measured with anti-IgG. PF3D7_0823200 was the most abundant protein detected, and 94 additional proteins were significantly co-purified (Fig. 6b and Supplementary Data 7). GO enrichment analysis showed that the detected proteins are mainly involved in translation, RNA splicing, and ribonucleoprotein complex biogenesis (Fig. 6c), confirming a role of PF3D7_0823200 in diverse RNA pathways. These candidates were cytoplasmic as well as nuclear, validating the presence of PF3D7_0823200 in both cellular compartments. Among these potential partners, we detected CUGBP Elav-like family (CELF) members 1 and 2 (PF3D7_1359400 and PF3D7_1409800,

respectively), which were also significantly shifted in the R-DeeP, supporting the hypothesis that PF3D7_0823200 may be associated with the CUGBP Elav-like family. To further support the interaction of these candidates with PF3D7_0823200, we calculated their correlation using the R-DeeP data. A good correlation (> 0.6) was observed for 52 of them, suggesting that they may form complex(es) with our bait (Supplementary Data 7). Although low correlations were obtained for CELF1 and 2 (−0.67 and 0.21, respectively), we can speculate that these proteins are components of several splicing complexes, hindering the Pearson's correlation coefficients.

### PF3D7_0823200 is an RBP regulating *var* and *ap2* transcripts
Based on its interactome and predicted function, we sought to identify the RNA targeted by PF3D7_0823200 and performed an eCLIP-seq experiment[17,36]. Briefly, RNA-protein complexes were UV-crosslinked and immunoprecipitated, and RNAs were reverse-transcribed for high-throughput sequencing. eCLIP-seq experiments were performed in duplicate and anti-IgG was used as negative control. Using Piranha, a CLIP-seq peak caller[37], and stringent filters, we detected a total of 512 peaks after comparison of PF3D7_0823200 IP and Input samples while

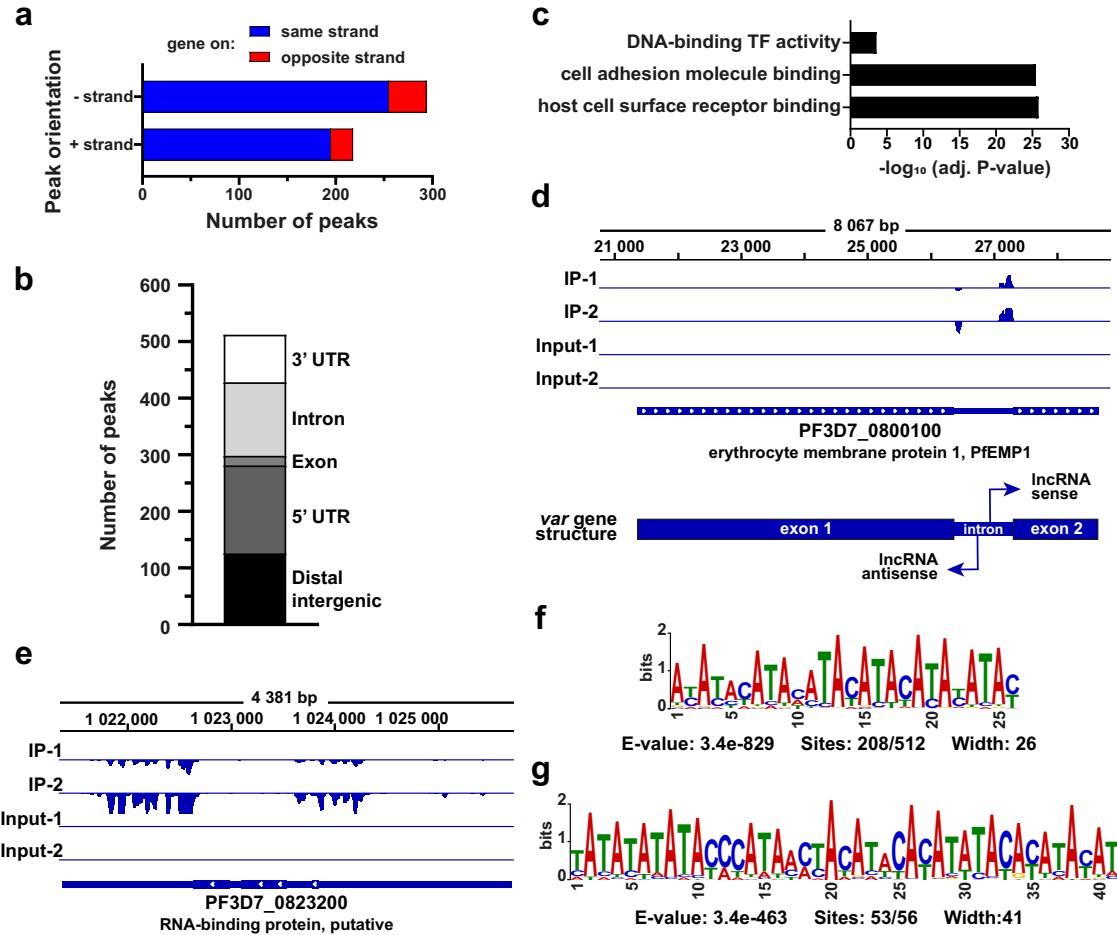

**Fig. 7 | Identification of PF3D7_0823200 targets using eCLIP-seq. a** Number and orientation of the peaks detected using Piranha. **b** Distribution of the peaks identified. **c** GO enrichment analysis of the significantly enriched transcripts. The top 3 terms of Molecular Function are represented as −log$_{10}$ (adjusted *P*-value) (Fisher's exact test with Bonferroni adjustment). **d**, **e** Tracks showing the eCLIP-seq peaks spanning on the region of the *var* gene, PF3D7_0800100 (**d**), and PF3D7_0823200 (**e**). The scales are −100-100 and −150-0, respectively. **f** Sequence logo of the most significant motif identified by MEME Suite search using the 512 eCLIP-seq peaks. **g** Sequence logo of the most significant motifs using the 56 peaks mapping on *var* genes. All significant motifs identified by MEME Suite search are represented in Supplementary Fig. 7.

only one peak was significantly identified with IgG samples (Supplementary Fig. 6 and Data 8). These peaks were distributed across 307 genes and 87.9% of the peaks were found in the same orientation as the associated genes (Fig. 7a). Only 17 peaks were detected on gene coding regions indicating a preferential binding of PF3D7_0823200 to untranslated RNA sequences. Indeed, 126 peaks were significantly identified on distal intergenic (24.6%), 155 on 5′ UTR (30.3%), 130 on intron (25.4%) and 84 on 3′UTR (16.4%) (Fig. 7b) suggesting that this RBP may play a role in RNA stabilization and/or splicing. GO enrichment analysis showed that these RNAs were associated with host cell surface, cell adhesion binding and DNA-binding transcription factor activity (Fig. 7c). Closer inspection revealed a specific interaction pattern for *var* genes with 56 significant peaks and 12 AP2 transcription factors, for which 44 peaks were called, including AP2-EXP, AP2-G5 and AP2-L (Fig. 7d and Supplementary Data 8). These two protein families were uniquely responsible for the observed GO enrichment. Although not all *var* peaks were called, we noticed a particular pattern with a first peak mapping at the start of the intronic region in the opposite orientation to the gene, while a larger second peak was detected at the end of the intron in the gene orientation (Fig. 7d). This arrangement is identical to the orientation and position of the well-studied sense and antisense lncRNAs of *var* genes that are transcribed from the bidirectional intron promoter[38–40]. Although further experiments are required to validate this finding, this result suggests that PF3D7_0823200

interact with *var* lncRNAs and participate in the regulation of the *var* gene family and their mutually exclusive expression[38–43]. For the AP2 transcription factors, the majority of the peaks were identified on the distal intergenic and 5′ UTR regions indicating that the protein interacts upstream of the AP2 coding regions. The detection of intergenic sequences could either reveal incorrect UTR annotations or the presence of ncRNAs and upstream ORF (uORF). In any case, this RBP seems to contribute to the dynamic balance observed between these master transcription factors. Similarly, we detected five peaks mapping the region of *gdv1*, an activator of sexual commitment, but the opposite orientation of these peaks relative to the gene most likely indicated that PF3D7_0823200 interacted with the antisense lncRNA. This lncRNA is described to inhibit *gdv1* transcription, thereby maintaining the gametocyte specific transcription factor, *ap2-g*, in a repressed state. Finally, the interaction of PF3D7_0823200 with its own 5′ and 3′ UTRs indicates a potential feedback loop that will need to be further validated at the experimental level to determine a positive or negative feedback mechanism (Fig. 7e).

Ultimately, the sequences of the 512 peaks were used to search enriched motifs using MEME Suite tool[44]. Although we found six enriched motifs with E-value < 0.05 (Supplementary Fig. 7a), the motifs 2-6 were not considered biologically meaningful because of their AT-richness (< 10%) or their low detection. The most significant motif was identified in 208 peaks (40%) and showed an E-value at 3.4e-829

(Fig. 7f). The 56 peaks mapping *var* genes were also analyzed separately and four motifs were significantly detected (Supplementary Fig. 7b). One particular motif was identified in 53 out of 56 sequences indicating its high specificity (E-value = 3.4e-463) (Fig. 7g). Collectively, these eCLIP-seq results confirmed that PF3D7_0823200 is a true RBP interacting with untranslated regions of various transcripts, known to control pathogenicity and sexual differentiation.

## Discussion

Our work provides the first proteome-wide screening of RDPs and RNP complexes in *P. falciparum*. Using the R-DeeP methodology, we identified 898 RDPs for which 39% were already associated with RNA. All techniques have their advantages and limitations, and the R-DeeP method is not an exception. Hence, R-DeeP is unique because it does not produce any potential enrichment bias, usually present in RBP screens, and identifies both proteins interacting directly and indirectly with RNA[20,45]. The approach also provides quantitative information about the RNA-dependence of protein and complexes. Limitations of R-DeeP are restricted detection of known RBPs interacting alone or in small complexes with RNA as well as those whose RNA is not critical for stability/formation of the RNP complex(es). Due to the R-DeeP experimental design, weak or transient protein-protein interactions may not be preserved during the ultracentrifugation step leading to enrichment of strong binding[20,45]. Overall, the generation of RNA interactomes with high confidence in malaria parasites would substantially benefit from the use of orthogonal and complementary strategies.

For the newly identified RDP candidates, a large proportion of them (181/545) corresponded to uncharacterized conserved or *Plasmodium*-specific proteins. These proteins may not directly interact with RNA but be part of protein complexes that are RNA-dependent, explaining why they were not previously classified as RBPs. Interestingly, 14 AP2 transcription factors were significantly impacted by RNase treatment indicating that they form RNP complexes. Among them, we detected AP2-G, AP2-G2 and AP2-G5, three major transcriptional regulators in the transition from asexual to sexual programs[46]. Although AP2-G expression is under the control of gdv1-lncRNA[47], no direct association has been described to our knowledge between the AP2-G transcriptional complex and (lnc)RNAs. Furthermore, AP2-HC is associated with heterochromatin[48], while AP2-SIP2 and AP2Tel bind to subtelomeric regions of *P. falciparum*[49,50]. These regions are known to be enriched in heterochromatin marks and lncRNAs such as lncRNA-TAREs, involved in telomere maintenance[51]. Although AP2-SIP2 was above our statistical threshold, unlike AP2-HC and AP2Tel, models have advanced potential synergies between AP2-SIP2 and lncRNA-TAREs for transcriptional regulation and recruitment of heterochromatin components[51]. Recruitment, assembly and regulation of these transcription factors may require the participation of lncRNAs to fulfill their respective biological functions.

The R-DeeP method also provided a snapshot of *Plasmodium* protein-protein interactions in presence and absence of RNA. As expected, a large majority of the proteins (76%) were in complex in the control condition. This percentage is most likely higher in cellular condition since all DNA-dependent complexes were affected by our DNase treatment in both experimental conditions. After RNase digestion, 13.2% of these proteins were no longer in complex indicating that the interactions of these proteins were RNA-dependent.

Co-segregation analysis of our R-DeeP data allowed us to reconstruct multiprotein complexes conserved in eukaryotes or specific of *Plasmodium*. A total of 131 protein complexes or clusters had a mean correlation > 0.6 suggesting that they were most likely detected in complex in our R-DeeP. Thus, this approach could be a useful tool for the malaria community to validate protein complexes in *P. falciparum* identified by various techniques such as in silico analysis[28,29], immunoprecipitation followed by mass spectrometry[17,31,52,53] and cryo-electron microscopy[54,55].

Our analysis also evaluated the RNA-dependence of these multi-protein complexes to discriminate which ones are ribonucleoprotein particles. As expected, the majority of spliceosomal complexes were unstable after RNase treatment as well as RNA-polymerase I-III complexes, while the integrity of 20 S proteasome and mitochondrial complexes were not sensitive to RNase activity.

Identified in our R-DeeP experiment, in silico analysis of PF3D7_0823200 suggested that this protein is well conserved in *Plasmodium* and shares similarities with CUGBP Elav-like family (CELF). This observation was supported by our IP-MS experiment showing a significant enrichment of translational and splicing factors, including the parasite CELF1 and 2. In the human nucleus, CELF proteins are involved in alternative splicing of pre-mRNA and RNA editing, while in the cytoplasm, they are associated with mature mRNAs and participate in deadenylation, stability and translation of the transcripts[33,56]. Several studies demonstrated that CELFs expression can be dysregulated by miRNAs and lncRNAs and are associated with development of human cancers[56]. Reciprocally, CELF proteins also have cooperative or antagonistic roles on ncRNA expression and function. In view of the nucleo-cytoplasmic localization and eCLIP-seq results, it is tempting to consider that PF3D7_0823200 could also be involved in similar processes. Indeed, the large majority of the eCLIP-seq peaks were detected in UTR and intronic regions suggesting a potential role in mRNA stabilization and/or RNA splicing. Although PF3D7_0823200 appeared to interact with a variety of transcripts, an enrichment was observed for *var* genes involved in antigenic switching and immune evasion, as well as AP2 transcription factors, master regulators of stage conversion. Interestingly, this protein was also associated with ncRNA transcripts, including the antisense lncRNA of *var* genes and *gdv1*, emphasizing its fundamental role regulating RNA metabolism in *P. falciparum*.

Recently, a transposon mutagenesis screening in *P. falciparum* indicated that disruption of PF3D7_0823200 impaired development of asexual stages with a mutant fitness score of −2.165[57]. Similarly, knockout mutations in *P. berghei* showed that its ortholog, PBANKA_0707400, was also severely impacted during blood stages and transitions to oocyst and sporozoite[58,59]. We can postulate that the role of PF3D7_0823200 in RNA regulation is fundamental for the parasite survival, but further functional and phenotypic assays using CRISPR-Cas9 transgenic lines will be required. As a whole our data indicate the power of our R-DeeP approach to identify novel RBPs and RDPs and contribute to explore multiprotein complexes and their RNA-dependence in *Plasmodium*.

## Methods

### Parasite lysate preparation and RNase treatment

Cultures of *P. falciparum* NF54 were synchronized by D-sorbitol treatments and $5 \times 10^{10}$ late trophozoites were treated with 0.15% saponin. After PBS washing, parasites were evenly separated into 2 samples (Control and RNase) and suspended in lysis buffer (25 mM Tris-HCl pH 7.4, 150 mM KCl, 0.5% (v/v) Igepal CA-630, 2 mM EDTA, 0.5 mM DTT, 1X protease inhibitor cocktail (Roche, 04693159001), 1X phosphatase inhibitor (Roche, 04906837001) and for Control sample only: 1200 units of RiboLock RNase Inhibitor (Thermo Scientific, EO0381)). After 30 min of incubation on ice with vortexing every 5 min, a freeze-thaw followed by homogenization using a 26 ½ G needle was performed on the parasite extract to improve the lysis efficiency. This step was repeated for a total of 3 freeze-thaw cycles. The soluble protein extract was obtained after centrifugation at $17,000 \times g$ for 15 min at 4 °C. Subsequently, the Control sample was treated with 100 units of DNase I (NEB, M0303) and 1X DNase I reaction buffer for 1 h at room temperature, while the RNase sample was treated with 100 units of DNase I, 500 units of RNase I (Ambion, AM2294), 50 units of RNase H (NEB, M0297), 200 µg of RNase A (Invitrogen, 12091021) and 1X DNase I reaction buffer. During the incubation, protein concentrations were quantified by Bradford assay (Sigma-Aldrich, B6916). The quality of the

enzymatic treatments was assessed on 1.2% agarose gel after phenol:chloroform:isoamyl alcohol (25:24:1, v/v) purification (Supplementary Fig. 1a).

## Sucrose density gradient preparation and ultracentrifugation

Ten sucrose solutions from 50% to 5% (w/v) sucrose were prepared in 10 mM Tris (pH 7.5), 1 mM EDTA (pH 8) and 100 mM NaCl as previously described[20]. First, 1 mL of the 50% sucrose solution was added to the bottom of the tube (Beckman Coulter, 344059) and flash frozen in liquid nitrogen. Then each sucrose solution was layered on top of the previous solution and frozen prior to addition of the next layer with the 5% sucrose solution on top of the tube. The sucrose density gradients were stored at −20 °C and thawed slowly on ice before adding protein lysates.

The Control and RNase samples were carefully overlaid on top of the thawed sucrose gradients avoiding any disturbance. For each condition and replicate, 2 to 2.5 mg of proteins were loaded onto the sucrose gradients. Ultracentrifugation was performed in Beckman L8-70M Ultracentrifuge equipped with a SW 41 Ti Swinging-Bucket Rotor (Beckman Coulter, 331362) at 110,000 x g for 18 h at 4 °C. After centrifugation, 25 fractions (~440 μL each) were carefully transferred by pipetting into fresh 1.5 mL tubes. Fraction 1 corresponded to the top of the tube and fraction 25 to the bottom. The different fractions were stored at −80 °C for western blot analysis or precipitated with 20% trichloroacetic acid (TCA) for mass spectrometry.

## Preparation of samples for mass spectrometry

TCA precipitated samples were resuspended in 30 μL buffer containing 100 mM Tris-HCl, pH 8.5 and 8 M urea. Disulfide bridges were reduced with tris(2-carboxyethyl)phosphine (5 mM final concentration) for 30 min at room temperature. Free SH groups were alkylated with chloroacetamide (CAM, 10 mM final concentration) for 30 min at room temperature in the dark. Proteins were first digested with 0.1 μg endoproteinase Lys-C for 6 h at 37 °C. Samples were then diluted with 100 mM Tris-HCl, pH 8.5 to reduce the concentration of urea to 2 M, $CaCl_2$ was added (2 mM final concentration), and digestion was continued with the addition of 0.5 μg trypsin. Samples were incubated at 37 °C overnight with shaking and reactions were quenched with the addition of formic acid (5% final concentration).

## Mass spectrometry analysis

Each sample was loaded onto a split triple-phase fused silica microcapillary column prepared as described previously[60]. Peptides were eluted from the column using a series of 10 - 2 h MudPIT steps. Mass spectrometry was performed using an Orbitrap Elite Hybrid mass spectrometer in positive ion mode.

## Peptide detection and quantification

Mass spectrometry data was generated, in two replicates, from 25 sucrose gradient fractions in both Control and RNase. A total of 1108 runs were carried out, and each raw file was converted to mzML format using the msconvert command in Proteowizard[61] using default parameters.

Proteins were detected and quantified separately in each fraction using Crux version 3.2-46bb0c1[62], in four steps. First, the canonical *Plasmodium* and human reference proteomes were downloaded from PlasmoDB (release 47)[22] and UniProt (UP000005640), respectively, and concatenated into a single FASTA file. A peptide index was created using the tide-index command, requiring fully tryptic peptides, up to two missed cleavages, and up to three methionine oxidations per peptide. These settings yielded a total of 5,669,753 distinct tryptic peptides. For each target peptide, a corresponding shuffled peptide decoy was also stored in the index. Second, the Tide search engine (tide-search command in Crux) was used to search spectra from each fraction against the Tide index. The search employed the exact p-value score function[63],

allowing isotope errors of 1 or 2 m/z, using an m/z bin width of 1.0005079 m/z, and using Param-Medic[64] to automatically select an appropriate precursor window size. Other Tide parameters were left at their default values. Third, all of the resulting peptide-spectrum matches (PSMs) from each fraction were analyzed jointly using Percolator[65], also via Crux, using default parameters. This step yielded, for each fraction, a list of proteins with associated q-values. Fourth, the Crux spectral-counts command was used to compute a normalized spectral abundance factor (NSAF)[66] for each protein in each fraction. In the NSAF calculation, only peptides identified at 1% peptide-level FDR by Percolator were considered. The NSAF values were aggregated into a set of matrices, one per replicate and treatment, in which rows are proteins and columns are sucrose gradient fractions. Proteins with no corresponding PSMs in a given fraction receive an NSAF value of zero. The total number of distinct proteins identified in at least one of the four settings (treatment or control, two replicates) was 5214.

## R-DeeP: statistical analysis

Prior to statistical analysis of the four NSAF matrices, three preprocessing steps were performed. First, protein quantifications were averaged across replicates, yielding one matrix for control and one for RNase treatment. We required that each protein have at least two consecutive non-zero average NSAF values in both treatment and control. Proteins that failed this criterion for either treatment or control were eliminated from both matrices. This step reduced the number of rows in each matrix from 4,146 to 3,671. Second, each matrix row was normalized to have a total abundance of 1. Third, each row was converted to a cumulative density, so that the value at row i and column j is the proportion of the abundance associated with protein i that is observed at or before fraction j. Given this preprocessed matrix, we use a Wilcoxon rank-sum test to detect proteins that exhibit a statistically significant shift in NSAF values between treatment and control. The statistic is based on the first 24 entries in each row, since the 25th entry is 1 by definition. Note that this step is equivalent to computing the area under a receiver operating characteristic (ROC) curve between the two distributions. In our setting, we are only interested in proteins for which the RNase peak comes before the control peak; hence, we use a one-tailed test in which values > 0.5 correspond to shifts in the desired direction. Finally, we subject the Wilcoxon p-values to FDR control using the Benjamini-Hochberg procedure[67].

GO enrichment analyses were generated using PlasmoDB (Fisher's exact test with Bonferroni adjustment).

## Comparison of RNA-associated protein datasets

A list of *Plasmodium* RNA-associated proteins was produced by collecting the datasets of the two in silico studies[18,19] and the mRNA proteome capture experiment[19]. In addition, we also integrated proteins annotated on PlasmoDB[22] with GO terms associated with RNA (GO:0016071: mRNA metabolic process; GO:0140098 catalytic activity, acting on RNA; GO:0006396: RNA processing; GO:0003723: RNA binding; GO:0005840: Ribosome). This final collection of 1,319 unique RNA-associated proteins is described in Supplementary Data 2 and was compared to the 898 significant proteins provided by this R-DeeP experiment. The Venn diagrams were generated using DeepVenn[68] and the UpSet plot was generated using UpSetR[69].

To investigate the interactome of the 545 proteins unique to the R-DeeP dataset, we reported the cluster(s) and potential partners from a protein-protein study[23]. These partners were then compared to our list of RNA-associated proteins as well as the 898 significantly shifted proteins from our R-DeeP experiment. The Enzyme Commission (EC) numbers were also reported when available from PlasmoDB[22] and OrthoMCL[24]. Then, all orthologs for each EC cluster were compared to RBP2GO, a database reported 22,552 RBP candidates from 13 different species[25]. All information is reported in Supplementary Data 2.

## Custom antibody production and purification

The different candidates were selected based on their shifting rank, molecular weight and (un)known function. The list included 6 proteins significantly shifted: PF3D7_0528600, PF3D7_1354900, PF3D7_1360100, PF3D7_0823200, PF3D7_0916700 and PF3D7_1347500, and two negative controls: PF3D7_1353900 and PF3D7_1465000. Peptide antigens were designed to target the C-terminal region and are indicated in Supplementary Data 3. They were used to immunize two rabbits and antisera from day 72 post-immunization were collected (Thermo Fisher Scientific). Antibody specificity was tested by western blot analysis on total *P. falciparum* protein extract. For each protein, the best antiserum was affinity-purified (Thermo Fisher Scientific) and validated by western blot analysis (Supplementary Fig. 2a).

## Western blot analysis

For each fraction, 27 µL (6% of total volume) was suspended with 8 µL of 4x Laemmli Sample Buffer (BioRad, 1610747). The samples were boiled at 95 °C for 5 min and loaded to 10% polyacrylamide gel. After migration, proteins were transferred onto a PVDF membrane using a Trans-Blot SD Semi-Dry Transfer Cell (BioRad) at 15 V for 30 min. Then the different membranes were blocked for 1 h at room temperature in WesternBreeze™ Solution (Invitrogen, WB7050) and incubated with the respective primary antibody (1:50 in WesternBreeze; 1:10,000 for anti-Aldolase) at 4 °C overnight with regular shaking. After 3 washes with WesternBreeze™ Wash Solution (Invitrogen, 46-7005), the blots were probed with HRP-labeled Goat anti-Rabbit IgG (H + L) (1:10,000, Novex™, A16104). Next, Clarity™ Western ECL Substrate (Bio-Rad, 1705060) was applied to develop the membranes. For each antibody, all 50 fractions distributed over four membranes (25 fractions per condition) were analyzed simultaneously by a ChemiDoc™ (BioRad). Relative protein abundance was normalized using Image Lab software (Bio-Rad).

## Molecular weight analysis

For each protein, the halfway was calculated and corresponded to the value for which the CDF reached 0.5, indicating that 50% of the total protein amount was detected. The apparent MW was obtained using the reference extrapolation ($y = 1146.9x^{2.2577}$; $R^2 = 0.9984$) based on position and molecular weight of reference human proteins[20]. Proteins were filtered by setting a cut-off of 0.5 and 2 for the apparent MW/theoretical MW ratio. Each protein was classified as smaller (<0.5), monomeric (0.5 < x < 2), larger (<2) or precipitated (hallway ≥ 24).

## Analysis of random, *Plasmodium* complexes and protein clusters

To compare Pearson correlation and treeClust[26], we labeled all pairs of proteins for which both proteins are in the complex as positives and all pairs for which one protein is in and the other not in the complex as negatives. Then we ranked pairs of proteins using two different scores: the Pearson correlation and treeClust dissimilarity score. For each ranking, we generated one ROC curve for each of 487 protein complexes, and we computed the area under each curve (AUROC). For the set of *Plasmodium* proteins, the average AUROC was 0.72 for Pearson correlation and 0.73 for treeClust. The corresponding values for the Hillier set were 0.62 (Pearson) and 0.61 (treeClust). Moreover, the AUROC values for the two methods were highly correlated across complexes: 0.95 for the *Plasmodium* set and 0.82 for the Hillier set (Supplementary Fig. 3) and showed no significant difference according to the signed-rank test ($p = 0.77$). PCA was generated using sklearn.decomposition package v1.3.2 in Python. Proteins were randomly distributed within each random complex. Three complexes were generated for each number of individual proteins (2 to 15). The different *Plasmodium* complexes were selected from various publications: Pf60S and Pf40S[54], U1, U2, U2-related, SF3a, SF3b, U5, U4-U6, snRNP, U6 (LSm), tri-snRNP, Prp19-CDC5, non-snRNP, NMD and SR-nRNP[28], Invasion-AMA1[70,71], IMC and Glideosome[72], PTEX[53], RAP[17], RNA-exosome[73], Mitochondrial complexes[74], Mediator-complex[31], SIP2[49], PfAP2Tel[50], DOZI-eIF4E[75], PfHSP40 and PfHSP70x[52], Kae1api[76], 20S-Proteasome and 26S-Proteasome[29], RhopH[55], Basal-complex[77], and CCR4-NOT, AMA1-MSP-RON, mRNA-decapping, RNA-polymerase complexes[71]. The protein clusters were extracted from a large-scale protein interactome study[23]. The protein composition of all complexes and clusters investigated are indicated in Supplementary Data 5.

For each pair of proteins within a complex, we computed the correlation of the proteins' normalized quantification profiles. These correlations were averaged across all non-identical pairs within the complex, and the associated *p*-values (one-sided *t*-test) were multiplied together and adjusted for multiple hypotheses using Fisher's method (Supplementary Data 5).

## Immunofluorescence assays

3D7 *P. falciparum* parasites were fixed with 4% paraformaldehyde and 0.0075% glutaraldehyde for 15 min at 4 °C and then sedimented on coverslips coated with Poly-L-ornithine (Sigma-Aldrich, P4957) for 1 h at room temperature. After two PBS washes, fixed parasites were permeabilized and blocked with 0.2% Triton X−100, 5% BSA, 0.1% Tween 20 in PBS for 30 min at room temperature. The custom anti- PF3D7_0823200 was diluted at 1:100 in PBS, 5% BSA, 0.1% Tween 20, and applied for 1 h at room temperature. After washing with PBS, parasites were incubated for 1 h with Donkey anti-Rabbit Alexa Fluor 568 (1:2000, Invitrogen, A10042). Slides were mounted in Vectashield Antifade Mounting Medium with DAPI (Vector Laboratories, H-1200). Images were acquired using a Keyence BZ-X810 fluorescence microscope and treated with ImageJ ($n > 30$ parasites).

## Immunoprecipitation followed by MudPIT mass spectrometry

A total of $7.5 \times 10^9$ 3D7 parasites enriched in late asexual stages were extracted by saponin lysis and resuspended in 50 mM Tris-HCl pH 7.5, 150 mM NaCl, 1% Triton X-100, 5 mM EDTA, 1 mM AEBSF and EDTA-free protease inhibitor cocktail (Roche, 11873580001). Soluble proteins were extracted using a 26 G needle and treated with 100 units of DNase I (NEB, M0303) for 10 min at room temperature. After centrifugation at 17,000 x g for 15 min at 4 °C, protein lysates were pre-cleared with Dynabeads™ Protein A (Invitrogen, 10001D) for 1 h at 4 °C. Our custom anti-PF3D7_0823200 (1:100) was added in the precleared supernatant and were incubated overnight at 4 °C. Purified Rabbit IgG was used in the same condition as negative control (1:100, MP Biomedicals, 0855944). Immunoprecipitations of antibody-protein complexes were performed using Dynabeads™ Protein A for 1 h at 4 °C. Subsequently, proteins were washed twice in PBS, 1% Triton X-100, 1 mM EDTA, once in PBS, 1% Triton X-100, 1 mM EDTA and 0.5 M NaCl, then twice in PBS, 1 mM EDTA. Before TCA-precipitation, proteins were eluted in 0.1 M glycine, pH 2.8 and neutralized with 2 M Tris-HCl, pH 8.0. Affinity and control purifications using anti-PF3D7_0823200 and anti-IgG, respectively, were performed in three independent experiments.

Samples were processed and analyzed by MudPIT mass spectrometry as described above. Resulting.raw files were processed using the in-house software package RAWDistiller v1.0 to generate.ms2 files. These were searched using the ProLuCID search engine against a database containing 5527 *P. falciparum* protein sequences, 36661 human protein sequences, sequences for 419 common contaminants, and shuffled versions of all the above sequences for estimating false discovery rates. A static modification of +57.02146 Da was used for cysteine residues (carbamidomethylation) and a variable modification of +15.9949 Da for methionine residues (oxidation). After searching, the resulting.sqt files were processed using DTASelect (v1.9)[78] using our in-house software swallow to select peptide spectrum matches

such that false discovery rates at the peptide and protein levels were less than 5%. Peptides and proteins from all samples were compared using Contrast[78], and dNSAF values calculated using our in-house software NSAF7 (v0.0.1). The statistical framework QPROT[79] was used to determine a subset of proteins enriched by the anti-PF3D7_0823200 antibody compared with negative controls (log2 fold change > 3, FDR < 0.01, mean dNSAF ≥ 0.0005).

**Enhanced crosslinking and immunoprecipitation followed by high-throughput sequencing (eCLIP-seq)**
A total of $7.5 \times 10^9$ 3D7 parasites enriched in late asexual stages were extracted by saponin lysis and crosslinked on ice by 254 nm UV light for a total of 1200 mJ/cm² with 2 min breaks using Spectrolinker™ XL-1000. The custom anti-PF3D7_0823200 or purified Rabbit IgG (15 µg) were coupled with Dynabeads™ M-280 Sheep Anti-Rabbit IgG (Thermo Fisher, 11203D) for 1 h at room temperature and then added to the parasite lysate. The following steps were processed using eCLIP Library Prep Kit (Eclipse BioInnovations, ECEK-0001) according to the manufacturer's instructions and as previously described[17]. Based on the molecular weight of PF3D7_0823200 (~32 kDa), regions of the nitrocellulose membrane ranging from 27 to 100 kDa were isolated and digested with proteinase K to release RNA. Libraries were generated by PCR amplifications consisting of 98 °C (30 s) followed by 6 cycles of (98 °C (15 s), 70 °C (30 s), 72 °C (40 s)), then 11 cycles of (98 °C (15 s), 72 °C (45 s)) and 72 °C (1 min). Library fragments of 175 to 350 bp were gel size-selected using MinElute Gel Extraction Kit (Qiagen, 28604) and the quantity and quality of the final libraries were assessed using a Bioanalyzer (Agilent Technology Inc). All samples were multiplexed and sequenced by dual indexed run (PE100) on the Illumina NovaSeq 6000 sequencer at the UC San Diego IGM Genomics Center and on the Illumina NextSeq 2000 Sequencing System at UC Riverside. The number of reads is reported in Supplementary Data 8. Bioinformatic analyzes were performed as previously described[17]. To normalize, all read counts were divided by the number of millions of mapped reads for each particular sample. Multi-mapping reads were conserved to avoid the loss of reads mapping to the repetitive *var* genes. Peak calling was performed using Piranha v1.2.1 with the options -z 50 (bin size 50), -l (convert covariates to log scale), the default q-value cutoff 0.01, and reads coverage ≥ 250. R package ChIPseeker v1.24.0 was used for peak annotation and was manually curated.

**Reporting summary**
Further information on research design is available in the Nature Portfolio Reporting Summary linked to this article.

## Data availability
Gene Ontology datasets used in this study can be accessed from PlasmoDB website (https://plasmodb.org/plasmo/app). eCLIP-seq datasets generated in this study have been deposited in the NCBI Sequence Read Archive under accession number PRJNA949221. The R-DeeP and IP-MS datasets have been deposited in the MassIVE repository with identification number MSV000091565 [https://massive.ucsd.edu/ProteoSAFe/dataset.jsp?task=c1704f223dda4177a932d7de7e7ea63e] and MSV000091228 [https://massive.ucsd.edu/ProteoSAFe/dataset.jsp?task=47d09718782e490ebaf90b0d66a744ea], respectively. Original data underlying this manuscript generated at the Stowers Institute can be accessed from the Stowers Original Data Repository (http://www.stowers.org/research/publications/LIBPB-2374).

## Code availability
The custom Python scripts used for eCLIP-seq analysis have been previously published[80]. The entire in-house software suite (Kite) used for the MudPIT mass spectrometry analysis is available in Zenodo (https://doi.org/10.5281/zenodo.5914885)[81].

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

## Acknowledgements
This work was supported by the National Institutes of Allergy and Infectious Diseases of the National Institutes of Health (grant R01 AI142743 to K.G.L.R.) and the University of California, Riverside (NIFAHatch-225935 to K.G.L.R.). This publication includes data generated at the UC San Diego IGM Genomics Center utilizing an Illumina NovaSeq 6000 that was purchased with funding from a National Institutes of Health SIG grant (#S10 OD026929).

## Author contributions
T.H., S.A. and K.G.L.R. conceived and designed all experiments. T.H. and S.A. performed the R-DeeP experiment. T.H. performed the western blots, microscopy imaging, protein immunoprecipitations, eCLIP-seq and data analysis. S.A. contributed to the eCLIP-seq data analysis. J.P. participated in the preparation and maintenance of cell cultures. C.B. and L.F. performed the mass spectrometry analyzes for the R-DeeP and IP-MS experiments. B.H., K.H. and W.S.N. contributed to the quantification and bioinformatic analysis of the R-DeeP data. T.H. and K.G.L.R. wrote the manuscript. All authors reviewed and approved the final manuscript.

## Competing interests
The authors declare no competing interests.
