## [Peer Review File · Nature Communications]

Proteome-Wide Identification of RNA-Dependent Proteins and
An Emerging Role for RNAs in Plasmodium falciparum Protein
ComplexesREVIEWER COMMENTS

Reviewer #1 (Remarks to the Author):

In the manuscript entitled "Proteome-Wide Identification of RNA-Dependent Proteins: An Emerging Role for RNAs in Plasmodium falciparum Protein Complexes", the authors Hollin et al. present a screen for RNA-dependent proteins (RDPs) in *P. falciparum* using the R-DeeP protocol that is based on sucrose density gradient ultracentrifugation followed by mass spectrometry analysis. They identified 463 proteins newly associated with RNA and further investigated protein-protein complexes and their relation to RNA. Finally, one candidate, PF3D7_0823200 is further investigated.

Altogether, the manuscript is well written and easy to read. The dataset generated as well as its analysis in terms of protein-protein interactions is very interesting and important to improve the characterisation and further elucidation of the role of RNA and RBPs/RDPs in *P. falciparum*. A better knowledge in this area can open new perspectives in the fight against malaria.

To my opinion, in order to strengthen the conclusions on PF3D7_0823200, it is essential to have an additional direct measurement of its binding to RNA. Major as well as minor comments and questions are listed in the following sections.

Major comments

- 1) To my opinion, the analysis focusing on PF3D7_0823200 suggests that the protein could be involved in the regulation of various (nc)transcripts, but it is not validated yet, so that the conclusions at this stage would need to be revised – essentially in the abstract and introduction.
- 2) In order to completely validate PF3D7_0823200 as RBP, it would be necessary to have either an image of the Protein-RNA complexes using a PNK assay and decreasing RNase concentration or a RAP for one of the RNA transcripts such as var lncRNA or from the AP2 transcription factors.
- 3) Supplementary data 5: how is the correlation calculated for complexes where only one protein was detected (ex: 2 subunits in the complex but 1 is missing. See NMD or Cluster255).
- 4) Supplementary Fig. 3: for 40S, it seems that some subunits are well correlated whereas some are not. Could you add a comment on this? Can it be that for large complexes, some subunits are "lost" during the ultracentrifugation step?
- 5) Line 253-254: it seems that most of the ribosomal proteins are not shifting the way it would be expected. Could the authors comment more on this result?
- 6) Lines 255-256 (p-value = 0.056, "complex is moderately disturbed by the RNase treatment") is in contradiction with lines 259-261 (p-value at 0.05, "did not appear to be RNA-dependent"). Clarify and update the conclusions.
- 7) Line 301: if the proteins PF3D7_1359400 and PF3D7_1409800 are potential partners of PF3D7_0823200, it would be interesting to document how the proteins correlate with each other's. Also, from all the potential interactors, what are the shifting proteins correlating best with PF3D7_1409800? Would interactions with these proteins functionally make sense?
- 8) Lines 345-351 and fig. 7f-g, supp fig 6a: the paragraph is confusing. The motifs seen in fig. 7 can also be considered as "AT" rich, which are not considered as "biologically meaningful". Please clarify why there is a particular attention given to the motifs by showing them in fig. 7?

Minor comments

- 9) Abstract line 21: what are the genetic assays performed in this study?
- 10) Introduction lines 32, 36 and 38, add references to validate the statements.
- 11) Supplementary Data 1: how were the four matrices further processed to compute the data presented in this table (one dataset for the control sample and one dataset for the RNase sample, but originally the data exists in duplicate)? How is the reduction to one dataset performed?
- 12) Fig. 2a: provide a supplementary table with the protein details for the three groups.
- 13) Lines 128-131: the explanation is too simplistic. There is another view presented in the discussion lines 406-409, which could help to revise the statement.
- 14) In addition to the Material and Methods section (line 538), the authors should make clear in the main text as well, why they do not comment/analyse the right shifted proteins.
- 15) Supplementary data 5: indicate in different color which complexes are considered as RNP complexes and which ones are not (see line 213).
- 16) Line 219: "Co-segregation analysis of LSm1 confirmed the LOWER correlation" instead of LOW correlation. According to line 221, a correlation of 0.673 (> 0.6) is a "good correlation".
- 17) Line 250: ref to Fig. 5d does not seem to be correct.
- 18) Line 262: add a reference to support the use of "confirming". Or, do the authors mean that an RNA dependence of this complex would not be expected? Please clarify.
- 19) Line 280: "implicated" instead of "implicating"?
- 20) Supplementary Data 7: add a column with information on the cellular localisation of the proteins or add reference in the text line 299.
- 21) Line 307: add ref for eCLIP.
- 22) Lines 324 to 329: add corresponding refs.
- 23) Line 359: add a ref validating the statement.
- 24) Line 367: add a ref for the AP2 and their functions.
- 25) Lines 388-389: add refs documenting the use of the different techniques in *P. falciparum*.
- 26) The discussion is a bit too long, so that the main messages get too diluted. Also, to my opinion the last sentence could emphasise again on the importance of the resource provided by this work and its meaning for the field.

Reviewer #2 (Remarks to the Author):

In the manuscript "Proteome-Wide Identification of RNA-Dependent Proteins: An Emerging Role for RNAs in *Plasmodium falciparum* Protein Complexes" Hollin et al. attempt to perform the first RNA-interacting protein complex study. In a more targeted section of the manuscript, the authors

characterise PF3D7_0823200 as an RNA-binding protein in *Plasmodium falciparum*. Despite using a proper set of experiments, that should have allowed the authors to solve the intended research questions, there are severe technical, analytical and interpretation issues that make this article unsuitable for publication.

1. The main experimental approach that the authors take is based on applying the R-DeeP method: An out of equilibrium density centrifugation approach where protein complexes sediment according to their physicochemical properties. The authors apply the method in the presence or absence of RNase treatment to evaluate if a protein complex is associated with RNA.

All conclusions from their R-DeeP experiments are based on $n=2$ replicates. Importantly, the authors consider it to be enough evidence to claim they have detected a protein when it appears in just one of the replicates: "values of zero in either matrix were ignored, so that the "average" of a non-zero value x in one matrix with a corresponding zero value in the other matrix was simply x ". So, effectively they can be drawing conclusions from $n=1$. There is no evaluation of how well the two replicates correlate, and therefore how robust the experimental approach is. There is neither a correlation of the 785 shifting proteins (per replicate). Do the 785 proteins consistently shift in both replicates?

2. In line 127, the authors say that they "noticed an enrichment of 142 ribosomal proteins". This is an unexpected high number of ribosomal proteins. How many proteins are there in *Plasmodium falciparum* ribosomes?

3. The authors find 463 proteins uncharacterised as RBPs but responding to RNase treatment. Do these proteins interact with other annotated RBPs? Authors could use protein-protein interaction studies, ideally in *Plasmodium falciparum*, but if not, they could use orthologue information, to evaluate if those proteins interact with known RBPs or if they may be part of a new RNA-interacting complex.

4. Their conclusion about the lack of RNase sensitivity of some well characterised RBPs is quite surprising, considering that they include ribosomal proteins, which can only form complexes through RNA interactions: "These data suggest that these particular protein complexes, although involved in RNA processes, are not RNP complexes or RNA-dependent for their formation and/or stability, at least in our experimental conditions".

5. There is no evaluation of the specificity of the antibodies. While the authors claim to have obtained antibodies for PF3D7_0823200, 151 PF3D7_1347500 and PF3D7_1353900 proteins, and that the antibodies "showed specific recognition" (line 151), there is no clear evidence that the antibodies recognise the target protein. The authors need to knock out/down the target protein to prove the specificity of the antibody.

6. Supplementary figures in their current form do not provide interpretable information:

Sup fig 2 is a PDF of 21 pages.

Sup fig 3 is a PDF of 48 pages.

Sup fig 4 is a PDF of 73 pages.

7. Regarding the protein-protein interaction networks and multiprotein sections (sections beginning in lines 164 and 204): Co-variation of proteins of the same complex does not mean that one can retrieve complex information. If all cellular protein was present just in one fraction, and absent in the next one, all proteins would have perfect covariance, but no information of specific complexes (or protein-protein interaction) could be retrieved. To determine if this approach can be used to interrogate protein complexes, the covariation needs to be unique to the proteins of each complex. There are formal approaches to analyse covariance. Authors should use an approach like the one of Kustatcher et al. *Nat Biotech* 2019. Moreover, the authors should represent their data in a comprehensive manner (instead of adding hundreds of pages of supplementary material as PDFs). PCA, tSNE or UMAP

representations would be a first step to assess if protein complexes are preserved.

8. No number of replicates is specified for the protein immunoprecipitation. Which is the number of replicates per condition (IgG and anti-PF3D7_0823200)?

9. At the end of the results section, authors say: "Collectively, these eCLIP-seq results confirmed that PF3D7_0823200 is a true RBP interacting with untranslated regions of various transcripts controlling pathogenicity and sexual differentiation." How can the authors confirm using eCLIP-seq that the RBP controls pathogenicity and sexual differentiation? As authors point it the discussion: "Recently, two large-scale genetic screening studies using piggyBac mutagenesis in *P. falciparum*²⁸ and barcoded *P. berghei* knockout mutations²⁹ were performed to identify essential genes in Plasmodium. Both studies indicated that disruption of PF3D7_0823200 or PBANKA_0707400 locus impaired development of asexual stages with a mutant fitness score of -2.165 in *P. falciparum* and a growth rate of 0.74 in *P. berghei*." Therefore, if the phenotypic characterisation is not part of this work, the authors should use their results to contextualise how the KO phenotype may be dependent on the RBP function of the protein they have discovered.

10. The manuscript is very poorly written: Authors consistently provide numbers of proteins or antibodies that are meaningless. They claim to have analysed 5214 unique proteins (line 97), but in the next sentence they mention that the number of proteins passing their own thresholds is 3671. The same is repeated when the authors claim having produced eight different custom rabbit polyclonal antibodies. Then, few lines later "only PF3D7_0823200, 151 PF3D7_1347500 and PF3D7_1353900 successfully showed specific recognition". Moreover, the manuscript lacks references such as (but not limited to): line 45 when discussing the number of RBPs in humans, line 47 when mentioning that Plasmodium falciparum is the deadliest malaria parasite, or line 48 when the authors say that RBPs mediate a wide range of essential processes in Plasmodium falciparum.

11. The authors claim having a good coverage of the parasite proteome (line 97 without giving any context. Which is the size of the Plasmodium falciparum proteome)?

12. Please, don't include speculations in the results section, e.g: "This discrepancy might be attributable to post-translational modifications only present when the protein is part of its RNP complex, but this requires further investigation." Moreover, if the agreement of one protein, according to the authors, validates the shifting of the MS data, the disagreement should invalidate it.

Reviewer #3 (Remarks to the Author):

The present study by Hollin et. al. investigates the abundance and complexity of a large-scale dataset obtained by a newly established proteome-wide technique, R-DeeP, to identify RNA-binding proteins (RBPs) as well as proteins in ribonucleoprotein complexes (RNPs) without direct RNA interaction. The authors took a rigorous approach which marries density gradient ultracentrifugation for fractionating and isolating RNPs followed by quantitative mass spectrometry. They eventually focus on a Plasmodium falciparum RNA-binding protein, PF3D7_0823200 and provide evidence using a CLIP-seq strategy to suggest that this protein may be, at least partly, responsible for co-transcriptional/post-transcriptional processing of transcripts important in malaria biology.

Taken together, the findings are convincing and presented clearly. This is a timely and important study in the malaria field, particularly at the time when extensive evidence is emerging on post-transcriptional and translational regulations that connect molecular mechanisms to pathogen virulence. Also, this study not only identifies novel plasmodium-specific RNA associated proteins but also validates numerous previously identified RNP protein candidates using a painstaking and

meticulous approach. The weak point is that the authors do not explain mechanistically how these RNA-associated proteins are relevant to biogenesis, stability/degradation or processing of these transcripts. Indeed, arguably these interactions might be indirect, and a detailed molecular mechanism might be beyond the scope of the study, some description of the nature of the transcriptional shut-down or inhibition of RNA processing could be included.

The manuscript warrants publication in Nature Communications but should have some conclusions rewritten or addressed. This reviewer has the following points for improving the manuscript, most of these have to do with clarity but in some cases the reviewer felt that there should be some validation experiments, if possible, to strengthen the conclusions-

1. The definition of RNA-dependent proteins (RDPs) is a relatively new concept. Typically, RNA-binding proteins (RBPs) binding to a RNA's surface of interaction depends on a diverse range of forces that are widely recognized. However, it is unclear what could be the recognition signal for protein-protein interactions of RBDs in the presence of RNAs. In many cases, RNP complex proteins can be important for chaperoning activity of RNA folding or for the stability of an RNA in the complex. To provide a broader, meaningful perspective, authors could provide evidence of RDB shifts and RNA stability/degradation when transcription is blocked or by attaining temperature-dependent RNP fractionation process (or cite existing evidence for the protein interactions of homologs which are validated).

2. When samples are not crosslinked, R-DeeP is expected to perform particularly well for stable RNA-protein interactions, and maybe less well for proteins with rapid on-off kinetics. By design, R-DeeP treats complexes as one unit and reflects RNA-binding activities of full RNPs rather than of individual subunits. In this respect, authors should clarify how they distinguish between binding events which are just casual vs mutually exclusive?

3. Page 6, Lines 159-162: "Interestingly, a difference of migration was observed for PF3D7_0823200.." – would this be possible if there is a change in the oligomerization status of the protein due to RNase treatment?

4. 'Fig. 2 Comparison of the significant left-shifted proteins'. Which cohort represents RDPs? please clarify in the figure legend.

5. The activity and aggregation of RBPs appear to be important in the context of protein translation. Macromolecular complexes containing RBPs, translational machinery, and mRNA transcripts consolidated to form granules through protein/protein interactions mediated by the glycine rich domains and protein/mRNA interactions. Given that translational repression plays an important role in *P. falciparum* gene expression, a short description on whether existence of such stress granule proteins are assessed for this interactome, would be recommended.

6. Some description in the section "RNA-dependence of Plasmodium ribonucleoprotein complexes" is not clear. CryoEM and biochemical analyses have shown that U6 RNA core allows structural rearrangements in the two catalytic steps of splicing for U4-U6.U5 tri-snRNP or U2-U6 complexes forming distinct RNP complexes, which are facilitated by a class of evolutionarily conserved RNA-dependent ATPases-helicases. How the U6 graph showing the mean correlation and number of proteins for all protein complexes investigated in Fig. 5b, containing mostly matchmaking type sm/Lsm proteins, relates with the distinct RNP clusters and spliceosome function?

7. Originally, the goal of R-DeeP technology was to define interactions of 'RNA-dependent proteins' or RDPs and as it appears, similarly authors have searched for and identified several RDPs in addition to known and unknown RBPs. Yet, they have experimentally characterized the protein PF3D7_0823200 which is a homolog/ortholog of a known RBP. To this end, one would wonder if characterizing a new RDP would have been more aligned with the characterization of malaria RNA interactome with R-DeeP

, a major focus of this manuscript? Also, authors please comment on whether RNA-centric methods such as poly-A capture or transcript-specific RBP affinity capture would be a more convincing way to characterize the interactions of RNA-associated proteins?

REVIEWER COMMENTS

Reviewer #1 (Remarks to the Author):

In the manuscript entitled “Proteome-Wide Identification of RNA-Dependent Proteins: An Emerging Role for RNAs in Plasmodium falciparum Protein Complexes”, the authors Hollin et al. present a screen for RNA-dependent proteins (RDPs) in *P. falciparum* using the R-DeeP protocol that is based on sucrose density gradient ultracentrifugation followed by mass spectrometry analysis. They identified 463 proteins newly associated with RNA and further investigated protein-protein complexes and their relation to RNA. Finally, one candidate, PF3D7_0823200 is further investigated.

Altogether, the manuscript is well written and easy to read. The dataset generated as well as its analysis in terms of protein-protein interactions is very interesting and important to improve the characterisation and further elucidation of the role of RNA and RBPs/RDPs in *P. falciparum*. A better knowledge in this area can open new perspectives in the fight against malaria.

To my opinion, in order to strengthen the conclusions on PF3D7_0823200, it is essential to have an additional direct measurement of its binding to RNA. Major as well as minor comments and questions are listed in the following sections.

We thank the reviewer for these overall positive comments on our manuscript. Our detailed answers to each comment are reported below.

Major comments

1) To my opinion, the analysis focusing on PF3D7_0823200 suggests that the protein could be involved in the regulation of various (nc)transcripts, but it is not validated yet, so that the conclusions at this stage would need to be revised – essentially in the abstract and introduction.

We have made changes in the revised version to address the reviewer comment.

2) In order to completely validate PF3D7_0823200 as RBP, it would be necessary to have either an image of the Protein-RNA complexes using a PNK assay and decreasing RNase concentration or a RAP for one of the RNA transcripts such as var lncRNA or from the AP2 transcription factors.

We would like to highlight that the eCLIP-seq is today the state-of-art technique to analyze RBP and provides sensitive and reproducible results to identify specifically RNA targets at the nucleotide resolution. Considering the presence of two RRM motifs on the protein as well as the results generated from our R-DeeP, IP-MS and eCLIP-seq experiments, we have no doubt that PF3D7_0823200 is a true RBP. We have however attempted to generate transgenic lines containing an epitope for the past year. While this line will help further characterization of this protein, transfection in *P. falciparum* remains problematic and laborious. So far, we have not been able to produce a genetically modified line for this specific protein.

To further validate that PF3D7_0823200 is interacting with RNA, we performed an HyPR-MS experiment in duplicate (PMID: 29972301) using either biotinylated probes targeting three AP2 transcription factors (9 probes total), or targeting var transcript/lncRNA in the sense and the antisense (3 probes for each). Unfortunately, PF3D7_0823200 was not detected in the immunoprecipitated samples. This protein was only detected at very low levels in our input samples, suggesting that this experiment is not suitable for this specific protein. Several RNase A protection assays were also tested where PF3D7_0823200 protein was immunoprecipitated, kept on magnetic beads, incubated with 10 ug of RNA and then incubated with RNase A as described in a published manuscript (see PMID: 31454137 for detail protocol). Detection of RNA was too low and in the absence of a positive control, we considered that this assay was inconclusive. We would like to indicate that in general RNase A assays used recombinant protein and not immunoprecipitated proteins that are more sensitive to RNA degradation. Moreover, due to the specific interaction of PF3D7_0823200 with intronic regions, we can hypothesize that the majority of the RNA is processed and matured leading to low binding events, which may not be visible on gel but can only be detected by sequencing. So far, the eCLIP-seq experiment that we generated successfully seems to be the most sensitive and specific experiment that we can think of to validate that PF3D7_0823200 is a true RBP.

3) Supplementary data 5: how is the correlation calculated for complexes where only one protein was detected (ex: 2 subunits in the complex but 1 is missing. See NMD or Cluster255).

For complexes in which only one protein was detected, the correlation was not calculated and 0 was indicated. For the NMD complex and cluster 255, 1 protein was missing. However, the number in the column “#proteins” indicated the number of proteins detected and not the total number of proteins. Thus, a total of three proteins are present in these complexes with 2 proteins detected and 1 missing. To avoid any confusion, the columns have been renamed as “#detected proteins” and “#missing proteins”. The correlation of complexes containing 0 or 1 detected proteins is now indicated as *nan*.

4) Supplementary Fig. 3: for 40S, it seems that some subunits are well correlated whereas some are not. Could you add a comment on this? Can it be that for large complexes, some subunits are “lost” during the ultracentrifugation step?

It is indeed highly possible that during the ultracentrifugation step some subunits are disassembled. We have included this hypothesis in the revised version of the manuscript.

5) Line 253-254: it seems that most of the ribosomal proteins are not shifting the way it would be expected. Could the authors comment more on this result?

Indeed, the ribosomal proteins were mainly not shifted in our R-DeeP. We think the RNA may be protected from RNase activity and only ribosomal proteins located on the surface of the complex could have been detached from the rest, explaining why some subunits of 40S are less correlated (previous comment). Despite that, the 40S was described as RNA-dependent (p-value=3.20E-05) while the 60S was just above the threshold (p-values at 0.056). We have modified and rearranged our conclusion about the lack of shift for ribosomal proteins.

6) Lines 255-256 (p-value = 0.056, “complex is moderately disturbed by the RNase treatment”) is in contradiction with lines 259-261 (p-value at 0.05, “did not appear to be RNA-dependent”). Clarify and update the conclusions.

We have corrected these conclusions in the revised version of the manuscript.

7) Line 301: if the proteins PF3D7_1359400 and PF3D7_1409800 are potential partners of PF3D7_0823200, it would be interesting to document how the proteins correlate with each other's. Also, from all the potential interactors, what are the shifting proteins correlating best with PF3D7_1409800? Would interactions with these proteins functionally make sense?

We thank the reviewer for this suggestion. We calculated the correlation between PF3D7_0823200 and its 94 potential partners. We observed that 49 proteins showed a good correlation (>0.6) with PF3D7_0823200, suggesting that these proteins can indeed form complex(es). For PF3D7_1359400 and PF3D7_1409800, low correlations were obtained (-0.67 and 0.21 , respectively) suggesting that these proteins may be part of several splicing complexes, hindering the Pearson's correlation coefficients. The statistics have now been added in the Results section and Supplementary Data 7.

8) Lines 345-351 and fig. 7f-g, supp fig 6a: the paragraph is confusing. The motifs seen in fig. 7 can also be considered as "AT" rich, which are not considered as "biologically meaningful". Please clarify why there is a particular attention given to the motifs by showing them in fig. 7?

As PF3D7_0823200 binds preferentially to intergenic and intronic regions, all identified motifs are indeed AT rich. We however paid particular attention to motif 1 for all peaks and motifs 1 and 2 for *var* peaks because they exhibited a higher GC content and/or were detected in the majority of the peaks. We have clarified this point in the revised manuscript and decided to only highlight the most significant motif for all and *var* peaks.

Minor comments

9) Abstract line 21: what are the genetic assays performed in this study?

We thank the reviewer for this comment. Genetic assays have been replaced by "molecular approaches".

10) Introduction lines 32, 36 and 38, add references to validate the statements.

References have been added to validate these statements.

11) Supplementary Data 1: how were the four matrices further processed to compute the data presented in this table (one dataset for the control sample and one dataset for the RNase sample, but originally the data exists in duplicate)? How is the reduction to one dataset performed?

This was done by averaging across replicates, while accounting for missing values, as described in lines 562-565 of the previously submitted manuscript: As biological replicates had an overall good correlation (Pearson correlation at 0.702), protein quantifications were averaged across replicates, yielding one matrix for control and one for RNase treatment. In this step only, values of zero in either matrix were ignored, so that the 'average' of a non-zero value x in one matrix with a corresponding zero value in the other matrix was simply x ."

12) Fig. 2a: provide a supplementary table with the protein details for the three groups.

The detailed list for the three groups is now included in the Supplemental Data 2.

13) Lines 128-131: the explanation is too simplistic. There is another view presented in the discussion lines 406-409, which could help to revise the statement.

We have revised this statement by merging both conclusions. To reduce the discussion section, this statement is now indicated in the result section.

14) In addition to the Material and Methods section (line 538), the authors should make clear in the main text as well, why they do not comment/analyse the right shifted proteins.

As indicated in L104-106, Gene Ontology enrichment analysis did not show any pathway corresponding to these proteins. The list of these proteins is available in Supplementary data 1.

15) Supplementary data 5: indicate in different color which complexes are considered as RNP complexes and which ones are not (see line 213).

The complexes are now highlighted based on their respective correlation (>0.6 and >0.8 are in blue and green, respectively). The significant RNase shift p-values are indicated in red.

16) Line 219: “Co-segregation analysis of LSm1 confirmed the LOWER correlation” instead of LOW correlation. According to line 221, a correlation of 0.673 (> 0.6) is a “good correlation”.

We agree with the reviewer that even if the U6-LSm1 complex still has a good correlation, this one is **lower** compared to the complex LSm2-8. We have modified the sentence accordingly.

17) Line 250: ref to Fig. 5d does not seem to be correct.

The correct reference (Fig. 5b) is now indicated.

18) Line 262: add a reference to support the use of “confirming”. Or, do the authors mean that an RNA dependence of this complex would not be expected? Please clarify.

Indeed, we didn't expect the mitochondrial complexes to be RNA-dependent. We have modified this sentence accordingly.

19) Line 280: “implicated” instead of “implicating”?

This has been replaced by “involved”.

20) Supplementary Data 7: add a column with information on the cellular localisation of the proteins or add reference in the text line 299.

This statement is based on the GO analysis showing a clear enrichment of cytoplasmic and nuclear proteins (Fig. 6c). A column has been added in the Supplementary Data 7 showing the Top 3 GO Cellular Components for the proteins significantly detected in the IP-MS.

21) Line 307: add ref for eCLIP.

References have been added.

22) Lines 324 to 329: add corresponding refs.

References have been added.

23) Line 359: add a ref validating the statement.

A reference has been added.

24) Line 367: add a ref for the AP2 and their functions.

A reference has been added.

25) Lines 388-389: add refs documenting the use of the different techniques in *P. falciparum*.

References have been added.

26) The discussion is a bit too long, so that the main messages get too diluted. Also, to my opinion the last sentence could emphasise again on the importance of the resource provided by this work and its meaning for the field.

We have reduced the discussion and highlighted the importance of the R-DeeP for the field in the last sentence.

Reviewer #2 (Remarks to the Author):

In the manuscript “Proteome-Wide Identification of RNA-Dependent Proteins: An Emerging Role for RNAs in Plasmodium falciparum Protein Complexes” Hollin et al. attempt to perform the first RNA-interacting protein complex study. In a more targeted section of the manuscript, the authors characterise PF3D7_0823200 as an RNA-binding protein in Plasmodium falciparum. Despite using a proper set of experiments, that should have allowed the authors to solve the intended research questions, there are severe technical, analytical and interpretation issues that make this article unsuitable for publication.

We thank the reviewer for these overall comments. We are sorry to hear that the reviewer found our article unsuitable for publication. We have addressed below the issues and concerns raised.

1. The main experimental approach that the authors take is based on applying the R-DeeP method: An out of equilibrium density centrifugation approach where protein complexes sediment according to their physicochemical properties. The authors apply the method in the presence or absence of RNase treatment to evaluate if a protein complex is associated with RNA.

All conclusions from their R-DeeP experiments are based on $n=2$ replicates. Importantly, the authors consider it to be enough evidence to claim they have detected a protein when it appears in just one of the replicates: “values of zero in either matrix were ignored, so that the “average” of a non-zero value x in one matrix with a corresponding zero value in the other matrix was simply x ”. So, effectively they can be drawing conclusions from $n=1$.

The reviewer is correct that we base our detection of proteins on single replicates. This is accomplished using standard techniques in the field of mass spectrometry: specifically, we search the observed spectrum against a database containing real (“target”) and shuffled (“decoy”) peptide sequences, and we use the rate of matches to these decoys as an estimate of the false discovery rate. The theory behind this target-decoy FDR estimation is well established (Elias 2007), and the Percolator software that we use to compute the FDR estimates is widely used in the field (Kall 2007). Overall, our approach is statistically conservative because we do not attempt to make use of the replicate structure of the data set to boost our statistical power to detect proteins.

Note that there is a second statistical question, which is not mentioned explicitly by the reviewer but is more closely related to our driving hypothesis. This question involves identifying proteins or sets of proteins that exhibit similar quantitative behavior across the entire collection of 25 fractions. For this question, we use $n=25$ (or, actually, $n=24$ because the CDF conversion effectively removes one observation) and we use a Wilcoxon rank sum test to identify significant shifts between the RNase and control fractions.

There is no evaluation of how well the two replicates correlate, and therefore how robust the experimental approach is. There is neither a correlation of the 785 shifting proteins (per replicate). Do the 785 proteins consistently shift in both replicates?

We agree that this is an important question. To answer it, we computed the shifting operation separately in the two replicates. Note that this approach necessarily yields fewer proteins in each of the two replicates.

In particular, applying the requirement that we observe two consecutive values per protein yields 3432 proteins in replicate 1 and 3148 proteins in replicate 2. These two sets share 3029 proteins in common, which is fewer than the 3671 that we obtained when analyzing the two replicates jointly. We next applied our statistical analysis separately to the two replicates, focusing on the 3029 proteins that were identified in both (Figure 1). Among the original set of 785 shifting proteins, only 595 appear with two consecutive non-zero NSAF values in both replicates. Among these, we observe that 56.5% shift significantly when analyzed separately in both replicates, and 585 proteins (98.3%) shift significantly in at least one of the two replicates. Overall, this analysis suggests that the two replicates are concordant, and that replicate 1 appears to yield better statistical power to detect shifts than replicate 2.

Figure 1: Consistency of shifts across replicates. The figure plots the statistical scores for the 3029 proteins that were identified in both replicates, with vertical and horizontal lines representing the 1% FDR threshold. A total of 367 proteins are significantly shifted in both replicates. The 595 proteins that shift significantly in the combined analysis are indicated in red.

2. In line 127, the authors say that they “noticed an enrichment of 142 ribosomal proteins”. This is an unexpected high number of ribosomal proteins. How many proteins are there in *Plasmodium falciparum* ribosomes?

In *Plasmodium falciparum*, 168 proteins are annotated as “ribosomal proteins”. This high number is due to the fact that the parasite has ribosome heterogeneity and specialization, with ribosomes expressed at different stages (A and S-type; PMID: 37053161) and localized in different subcellular compartments (nuclear/cytoplasm, mitochondria, apicoplast).

3. The authors find 463 proteins uncharacterised as RBPs but responding to RNase treatment. Do these proteins interact with other annotated RBPs? Authors could use protein-protein interaction studies, ideally in *Plasmodium falciparum*, but if not, they could use orthologue information, to evaluate if those proteins interact with known RBPs or if they may be part of a new RNA-interacting complex.

Protein-protein interaction studies are limited in *P. falciparum* and homology analysis is difficult due to the high A/T richness of the genome. However, we used a *Plasmodium* interactome study (PMID: 31390575) and noticed that 56/463 proteins interact either with known RBPs or other RDPs identified in this study. Moreover, we used OrthoMCL, a database grouping orthologous protein sequences, and reported the Enzyme Commission (EC) number for the 463 proteins (98 proteins were associated with EC numbers). Orthologs included in these EC clusters were compared to RBP2GO, a database grouping 22552 RBPs from 13 species. A total of 86 proteins had at least one ortholog previously identified as RBPs. All of this information is presented in Supplementary data 2 and summarized as a Venn diagram in Supplementary Fig. 1.

4. Their conclusion about the lack of RNase sensitivity of some well characterised RBPs is quite surprising, considering that they include ribosomal proteins, which can only form complexes through RNA interactions: “These data suggest that these particular protein complexes, although involved in RNA processes, are not RNP complexes or RNA-dependent for their formation and/or stability, at least in our experimental conditions”.

We have clarified the manuscript about the lack of RNase sensitivity for some well characterized RBPs. We highlighted that proteins can be involved in RNA pathways without being RNA-dependent to form stable complexes. These proteins are not part of a ribonucleoprotein complex but are still involved in RNA metabolism.

5. There is no evaluation of the specificity of the antibodies. While the authors claim to have obtained antibodies for PF3D7_0823200, 151 PF3D7_1347500 and PF3D7_1353900 proteins, and that the antibodies “showed specific recognition” (line 151), there is no clear evidence that the antibodies recognise the target protein. The authors need to knock out/down the target protein to prove the specificity of the antibody.

We agree with the reviewer's comment that knock out/down of the target protein is the best way to prove the specificity of custom antibodies. However, this type of validation is extremely difficult in *Plasmodium falciparum*. Indeed, RNAi machinery is absent in malaria parasites and therefore RNAi-based silencing methods have been shown to be ineffective. Thus, the generation of transgenic lines is necessary and transfections in *P. falciparum* are extremely challenging even with CRISPR-Cas9. Months and sometimes years are needed to obtain validated transgenic clones. This is why we excluded this validation method and generated custom antibodies. Instead, we verified the specificity of our custom antibodies by western-blot as presented in Fig. S2a. We only considered that three antibodies showed enough specific recognition to proceed with for R-DeeP confirmation. All of them showed similar western blot profiles compared to the MS profiles in the expected fractions as well as the theoretical protein size. Then, anti-PF3D7_0823200 antibody was used for IP-MS and the protein was the most abundant protein detected confirming the specificity of this custom antibody.

6. Supplementary figures in their current form do not provide interpretable information: Sup fig 2 is a PDF of 21 pages.

Sup fig 3 is a PDF of 48 pages.

Sup fig 4 is a PDF of 73 pages.

We are sorry to hear that the reviewer found the supplementary material to be uninterpretable. We were attempting to provide exhaustive information for readers who were interested in diving into the details related to a specific complex, but we agree with the reviewer that most readers will likely find this approach to be overwhelming. Accordingly, we revised the supplementary figures and only included the complexes and clusters discussed in the manuscript. The Sup fig 2 containing the random complexes was not conserved and Sup fig 3 and 4 (annotated as 4 and 5 in the revised manuscript) are now PDFs of 9 and 3 pages, respectively.

7. Regarding the protein-protein interaction networks and multiprotein sections (sections beginning in lines 164 and 204): Co-variation of proteins of the same complex does not mean that one can retrieve complex information. If all cellular protein was present just in one fraction, and absent in the next one, all proteins would have perfect covariance, but no information of specific complexes (or protein-protein interaction) could be retrieved. To determine if this approach can be used to interrogate protein complexes, the covariation needs to be unique to the proteins of each complex. There are formal approaches to analyse covariance. Authors should use an approach like the one of Kustatcher et al. *Nat Biotech* 2019. Moreover, the authors should represent their data in a comprehensive manner (instead of adding hundreds of pages of supplementary material as PDFs). PCA, tSNE or UMAP representations would be a first step to assess if protein complexes are preserved.

To address this comment, we first made a PCA representation of the protein quantifications for all proteins, color-coding those belonging to five *Plasmodium* complexes (Figure 2). Proteins that are part of the same complex cluster together, demonstrating that protein complex information is in fact well captured in our experiment.

Second, we compared the performance of two methods for detecting proteins belonging to the same complex: our original approach (based on the Pearson correlation between the protein quantification values) and the treeClust method used by Kustatcher et al. (*Nat Biotech* 2019), as suggested by the reviewer. Specifically, for each protein complex, we labeled all pairs of proteins for which both proteins are in the complex as positives and all pairs for which one protein is in and the other not in the complex as negatives. Then we ranked pairs of proteins using two different scores: the Pearson correlation used in our manuscript and the treeClust dissimilarity score. For each ranking, we generated one ROC curve for each of 487 protein complexes, and we computed the area under each curve (AUROC). In this analysis, we observed that the two scores performed very similarly: in *Plasmodium*, the average AUROC was 0.72 for both the Pearson correlation and treeClust. The corresponding values for the Hillier set were 0.62 (Pearson) and 0.61 (treeClust). Moreover, the AUROC values for the two methods were highly correlated across complexes: 0.95 for the *Plasmodium* set and 0.83 for the Hillier set (Figure 3) and showed no significant difference according to a signed-rank test ($p=0.76$). Figure 1c in the Kustatscher paper suggests that treeClust significantly outperforms Pearson correlation, but that result is in the context of predicting functional associations from Reactome, rather than co-complex membership. Also, the input to the Kustatscher

analysis represents quantification values from 294 distinct biological perturbations. This is quite different from, and much richer than, the 25 sucrose gradient fractions employed in our analysis. We conclude that, for the purposes of analyzing this dataset, Pearson correlation is sufficient, and the treeClust approach would not add value. We have included this additional information in the manuscript.

Figure 2: Clustering of proteins belonging to the same complex. We plot the 2D PCA representation of the protein quantifications for each protein, with colors corresponding to proteins belonging to five *Plasmodium* complexes with high correlation values. These proteins cluster together, indicating that our approach preserves protein complex information.

Figure 3: Comparison of protein similarity measures using Pearson correlation or treeClust dissimilarities. Each point corresponds to a protein complex, and the two axes represent AUROC values calculated using two different methods for ranking protein pairs.

8. No number of replicates is specified for the protein immunoprecipitation. Which is the number of replicates per condition (IgG and anti-PF3D7_0823200)?

The protein immunoprecipitation was performed in triplicate per condition as indicated in Figure 6 legend and Supplementary data 7. This information is also indicated in the Methods section.

9. At the end of the results section, authors say: “Collectively, these eCLIP-seq results confirmed that PF3D7_0823200 is a true RBP interacting with untranslated regions of various transcripts controlling pathogenicity and sexual differentiation.” How can the authors confirm using eCLIP-seq that the RBP controls pathogenicity and sexual differentiation? As authors point it the discussion: “Recently, two large-scale genetic screening studies using piggyBac mutagenesis in *P. falciparum*²⁸ and barcoded *P. berghei* knockout mutations²⁹ were performed to identify essential genes in Plasmodium. Both studies indicated that disruption of PF3D7_0823200 or PBANKA_0707400 locus impaired development of asexual stages with a mutant fitness score of -2.165 in *P. falciparum* and a growth rate of 0.74 in *P. berghei*.” Therefore, if the phenotypic characterisation is not part of this work, the authors should use their results to contextualise how the KO phenotype may be dependent on the RBP function of the protein they have discovered.

Indeed, the eCLIP-seq results do not confirm that the RBP controls pathogenicity and sexual differentiation. We wanted to indicate that the eCLIP-seq showed that the RBP interacts with transcripts and these transcripts are associated with these pathways. We have clarified this statement. For the essentiality of PF3D7_0823200, we thank the reviewer for his suggestion. We have shortened this section and contextualized our results with these phenotypic assays.

10. The manuscript is very poorly written: Authors consistently provide numbers of proteins or antibodies that are meaningless. They claim to have analysed 5214 unique proteins (line 97), but in the next sentence they mention that the number of proteins passing their own thresholds is 3671. The same is repeated when the authors claim having produced eight different custom rabbit polyclonal antibodies. Then, few lines later “only PF3D7_0823200, 151 PF3D7_1347500 and PF3D7_1353900 successfully showed specific recognition”. Moreover, the manuscript lacks references such as (but not limited to): line 45 when discussing the number of RBPs in humans, line 47 when mentioning that *Plasmodium falciparum* is the deadliest malaria parasite, or line 48 when the authors say that RBPs mediate a wide range of essential processes in *Plasmodium falciparum*.

We are sorry to hear that the reviewer found the manuscript to be poorly written. We agree that the number of unique proteins before filtering is dispensable and this information has been deleted in the revised manuscript. We indicated that we produced 8 different custom antibodies, but only three of them showed a specific recognition. We considered that this information is important as it clarifies that production of specific antibodies in *P. falciparum* is challenging and that we didn't just test 3 antibodies that all worked.

Regarding the lack of references, two reviews were cited in our original manuscript for the number of RBPs in humans and model organisms (references 9 and 10, previously annotated 4-5). To address the reviewer comment regarding the lack of citations, we have added several references throughout the revised version of the manuscript.

11. The authors claim having a good coverage of the parasite proteome (line 97 without giving any context. Which is the size of the *Plasmodium falciparum* proteome)?

The proteome of *P. falciparum* contains 5545 proteins and we have calculated the cumulative distribution function for 3671 proteins (66%). This information has been added to the revised version of the manuscript.

12. Please, don't include speculations in the results section, e.g: “This discrepancy might be attributable to post-translational modifications only present when the protein is part of its RNP complex, but this requires further investigation.” Moreover, if the agreement of one protein, according to the authors, validates the shifting of the MS data, the disagreement should invalidate it.

We have avoided as much as possible speculations in the results section. However, to reduce the discussion as requested by Reviewer 1, we had to keep some key hypotheses in the result section. We don't consider the shifting of the MS data due to the presence or absence of PTMs but to their shift detected after RNAase treatment. This hypothesis concerns the difference of migration in the western-blot analysis. PF3D7_0823200 was detected at a higher molecular weight in the RNase condition but still shifted to the left in the R-DeeP fractions, similarly to what was observed with the MS data.

Reviewer #3 (Remarks to the Author):

The present study by Hollin et. al. investigates the abundance and complexity of a large-scale dataset obtained by a newly established proteome-wide technique, R-DeeP, to identify RNA-binding proteins (RBPs) as well as proteins in ribonucleoprotein complexes (RNPs) without direct RNA interaction. The authors took a rigorous approach which marries density gradient ultracentrifugation for fractionating and isolating RNPs followed by quantitative mass spectrometry. They eventually focus on a *Plasmodium falciparum* RNA-binding protein, PF3D7_0823200 and provide evidence using a CLIP-seq strategy to suggest that this protein may be, at least partly, responsible for co-transcriptional/post-transcriptional processing of transcripts important in malaria biology.

Taken together, the findings are convincing and presented clearly. This is a timely and important study in the malaria field, particularly at the time when extensive evidence is emerging on post-transcriptional and translational regulations that connect molecular mechanisms to pathogen virulence. Also, this study not only identifies novel plasmodium-specific RNA associated proteins but also validates numerous previously identified RNP protein candidates using a painstaking and meticulous approach. The weak point is that the authors do not explain mechanistically how these RNA-associated proteins are relevant to biogenesis, stability/degradation or processing of these transcripts. Indeed, arguably these interactions might be indirect, and a detailed molecular mechanism might be beyond the scope of the study, some description of the nature of the transcriptional shut-down or inhibition of RNA processing could be included.

The manuscript warrants publication in Nature Communications but should have some conclusions rewritten or addressed. This reviewer has the following points for improving the manuscript, most of these have to do with clarity but in some cases the reviewer felt that there should be some validation experiments, if possible, to strengthen the conclusions-

We thank the reviewer for these positive comments on our manuscript. We have addressed all issues below and edited the manuscript accordingly.

1. The definition of RNA-dependent proteins (RDPs) is a relatively new concept. Typically, RNA-binding proteins (RBPs) binding to a RNA's surface of interaction depends on a diverse range of forces that are widely recognized. However, it is unclear what could be the recognition signal for protein-protein interactions of RBDs in the presence of RNAs. In many cases, RNP complex proteins can be important for chaperoning activity of RNA folding or for the stability of an RNA in the complex. To provide a broader, meaningful perspective, authors could provide evidence of RDB shifts and RNA stability/degradation when transcription is blocked or by attaining temperature-dependent RNP fractionation process (or cite existing evidence for the protein interactions of homologs which are validated).

Although we agree that studying the impact of transcription inhibition and temperature-dependence would provide interesting information for RDP shifts, RNA stability, and RNP rearrangements, we consider these experiments beyond the scope of this article. As RDP is a new concept, there are no studies available yet to our knowledge. We have however improved the introduction and conclusion to clarify this new concept and address concerns raised by the reviewers.

2. When samples are not crosslinked, R-DeeP is expected to perform particularly well for stable RNA-protein interactions, and maybe less well for proteins with rapid on-off kinetics. By design, R-DeeP treats complexes as one unit and reflects RNA-binding activities of full RNPs rather than of individual subunits. In this respect, authors should clarify how they distinguish between binding events which are just casual vs mutually exclusive?

We agree with the reviewer that R-DeeP leads to enrichment of strong interactions because weak or transient interactions may be lost during ultracentrifugation. We have added this information in the discussion. However, the R-DeeP method does not discriminate between interactions that may be casual or exclusive. Proteins involved in exclusive interactions should be more affected by RNase treatment if they are part of an RNP complex. And conversely, not be affected at all if the complex is not RNA-dependent. For proteins that may interact casually with various complexes, the shift may be more discrete and thus make the statistical test less significant. This is a limitation of the R-DeeP that cannot be changed.

3. Page 6, Lines 159-162: “Interestingly, a difference of migration was observed for PF3D7_0823200...” – would this be possible if there is a change in the oligomerization status of the protein due to RNase treatment?

We thank the reviewer for this hypothesis which has been included in the revised version of the manuscript.

4. ‘Fig. 2 Comparison of the significant left-shifted proteins’. Which cohort represents RDPs? please clarify in the figure legend.

All proteins identified as shifted in the R-DeeP experiment are RDPs by definition and only some of them are unknown or known RBPs. For the proteins unique to our R-DeeP experiment, it’s not possible to discriminate between unknown RBPs and RDPs. We have clarified this point in the legend.

5. The activity and aggregation of RBPs appear to be important in the context of protein translation. Macromolecular complexes containing RBPs, translational machinery, and mRNA transcripts consolidated to form granules through protein/protein interactions mediated by the glycine rich domains and protein/mRNA interactions. Given that translational repression plays an important role in *P. falciparum* gene expression, a short description on whether existence of such stress granule proteins are assessed for this interactome, would be recommended.

A short description has been added indicating that the majority of proteins associated with stress granules and P-bodies were identified as RDPs in our experiment.

6. Some description in the section “RNA-dependence of Plasmodium ribonucleoprotein complexes” is not clear. CryoEM and biochemical analyses have shown that U6 RNA core allows structural rearrangements in the two catalytic steps of splicing for U4-U6.U5 tri-snRNP or U2-U6 complexes forming distinct RNP complexes, which are facilitated by a class of evolutionarily conserved RNA-dependent ATPases-helicases. How the U6 graph showing the mean correlation and number of proteins for all protein complexes investigated in Fig. 5b, containing mostly matchmaking type sm/Lsm proteins, relates with the distinct RNP clusters and spliceosome function?

We have clarified the section “RNA-dependence of Plasmodium ribonucleoprotein complexes”. The U6 complex, containing LSm proteins, was based on a previous publication investigating the conservation of

spliceosomal complexes (PMID: 21245033). We have renamed this complex U6 (LSm). We agree that the different complexes presented in this study rearrange and interact with each other in a cellular context. But it was important to verify that these proteins correlated well in these different (sub)complexes. In this revised manuscript, we analyzed the correlation of the spliceosomal B complex, which includes U2, U4-U6, U5 complexes as well as the Sm and LSm proteins, among others. A total of 46 proteins were included and the mean correlation was only at 0.238 (Supplementary Fig. 4 and Data 5). We observed 4 distinct clusters, two of which were enriched with U4-U6 and U6 (LSm) proteins, and U2, U5, Sm and CAP proteins, respectively. Interestingly, the spliceosomal B complex showed the lowest RNase shift p-value ($2e-38$), confirming its RNA-dependence. We have described this complex in the result section of the revised manuscript.

7. Originally, the goal of R-Deep technology was to define interactions of 'RNA-dependent proteins' or RDPs and as it appears, similarly authors have searched for and identified several RDPs in addition to known and unknown RBPs. Yet, they have experimentally characterized the protein PF3D7_0823200 which is a homolog/ortholog of a known RBP. To this end, one would wonder if characterizing a new RDP would have been more aligned with the characterization of malaria RNA interactome with R-Deep, a major focus of this manuscript? Also, authors please comment on whether RNA-centric methods such as poly-A capture or transcript-specific RBP affinity capture would be a more convincing way to characterize the interactions of RNA-associated proteins?

We agree with the reviewer's comment that the characterization of a new RDP would be more interesting in this R-Deep study. With this goal, we selected for the production of custom antibodies, 2 negative controls (ApiCOX25, PSA7), 2 characterized RBPs (musashi, Alba4), 2 putative RBPs (PF3D7_0823200 and PF3D7_1360100), and also 2 unknown RDPs identified in this study (PF3D7_0528000 and PF3D7_1354900). However, among the putative and uncharacterized proteins, only the custom antibody against PF3D7_0823200 showed a specific recognition. It's also important to note that PF3D7_0823200 is only annotated as putative RBPs due to the presence of RRM motifs. This protein showed a limited homology with known RBPs and cannot be considered as a conserved RBP. Our results confirmed that this protein seems to play a role in the regulation of important *Plasmodium*-specific transcripts. In the absence of custom antibodies against novel RDPs, the only alternative is to add an epitope. Generation of some transgenic lines using CRISPR-Cas9 are in progress in the laboratory, but this process is long (months/years) and laborious in *Plasmodium*.

Each method has its advantages and disadvantages. Poly-A, affinity capture or crosslinking based methods may be more convincing to characterize only RBPs but they also introduce biases during the enrichment process. Here, the R-Deep allows us to identify not only RBPs and also provides information on the RNA-dependence of proteins and complexes, which is unique to this technique. We have included a small paragraph in the discussion section of the revised manuscript.

REVIEWER COMMENTS

Reviewer #1 (Remarks to the Author):

The new version of the manuscript has gain in clarity, which was already very good. The revision of the statements in the main text is well appreciated as well as the added supplementary information that increases transparency. The answers to the comments are satisfying and properly reflected in the revised manuscript. The study is of great interest for the field and adding value, especially when considering that the current number of RBP datasets for *P. falciparum* is very limited.

Minor comments:

- Line 154: space missing "of98"
- Figure 2: "Western blot" without hyphen
- Supplementary Data 5: in the provided version of the table, the complexes are not highlighted in color (see answer to point 18).

Reviewer #3 (Remarks to the Author):

In this revised manuscript by Hollin et. al., authors used a newer proteome-wide approach, called R-Deep, to characterize RNA dependent protein interactions in malaria parasite *Plasmodium falciparum*. Despite their central role in RNA function, the RNA-binding specificities of many RNA-dependent proteins remain unknown or incompletely defined in higher and lower eukaryotes. To address this, the authors have assembled a genome-scale collection of RDPs and RBPs and their RNA-binding domains (RBDs) and quantitatively assessed their specificities. This is a timely contribution from a research group with significant experience in malaria RNP analysis and will be a valuable contribution to the field. Validation of some of the findings could be a concern, however, genetic manipulations with *falciparum* is notoriously hard and time consuming and could take significant time to resolve. Overall, I'm satisfied with the revised version of the manuscript.

Reviewer #4 (Remarks to the Author):

The first question raised by the Reviewer 2 is indeed quite crucial since all the following quantitative analyses are strongly dependent on this first step. For the missing values, the authors effectively imputed the same values found in the other replicate. This seemingly simple calculation, however, represents many bold assumptions. Not being detected in one sample suggests low abundance and it is not clear how to correctly impute their abundance. When not established, I would suggest to take a rather conservative calculation, e.g., take the average. If taking average severely affects the whole analysis, it means that the whole finding is dependent on this bold imputation step. To clearly demonstrate the power of R-Deep, the author need to take average or even analyze each replicate separately and take intersection or union with a proper statistical validation.

On the other hand, the data analysis for peptide and protein identification was done in a very clear and rigorous way so I doubt that the proteins found contain many false positive identifications. For identification, taking union proteins from different replicates would not bring any issue as the FDR control has been done jointly using a well established method, which is Percolator. But quantification is another issue and imputation is one of the topics to which the field still struggles to find the correct answer.

Related to this, the Reviewer 2 raised the second question on the reproducibility between replicates.

Already the authors have the separate results from each replicate and could proceed with the follow up analysis. If the candidate PF3D7_0823200 was lost during this procedure, it suggests that this imperfect imputation was introducing the candidate. I agree that the candidate is a true RBP. But the authors could not argue strongly that their method was so powerful that this candidate has been found. I believe the method is one of the major points in this manuscript (while R-Deep already has been published) and should be robust for other applications. Even if the authors have to use a bit higher FDR (e.g., 3%) to find the valid candidates, I would recommend to do so since FDR is something that we know how to control while such a bold imputation is out of control.

Regarding the question 4: it is very interesting to see that the RNAase sensitivity varies for different RBPs. To address this, the authors suggested a hypothesis "we can hypothesize that in our 133 experimental conditions, the RNA may be protected from RNase activity and only ribosomal 134 proteins located on the surface of the complex could have been detached from the rest during 135 the RNase treatment." And then they report a possible limitation, which I believe raises the overall quality of the manuscript. However, I would recommend to give a few selected raw signals that show this new hypothesis is indeed the case. Since this is not only interesting and quite counter intuitive, I believe the authors should provide (at least) a few selected spectra that can help the readers to be convinced.

I believe other concerns raised by Reviewer 2 have been addressed.

REVIEWER COMMENTS

Reviewer #1 (Remarks to the Author):

The new version of the manuscript has gain in clarity, which was already very good. The revision of the statements in the main text is well appreciated as well as the added supplementary information that increases transparency. The answers to the comments are satisfying and properly reflected in the revised manuscript. The study is of great interest for the field and adding value, especially when considering that the current number of RBP datasets for *P. falciparum* is very limited.

We thank the reviewer for this kind comment.

Minor comments:

- Line 154: space missing “of98”

This has been corrected.

- Figure 2: “Western blot” without hyphen

The hyphen has been removed in the different figures.

- Supplementary Data 5: in the provided version of the table, the complexes are not highlighted in color (see answer to point 18).

We apologize for this mistake. The correct version is now uploaded.

Reviewer #3 (Remarks to the Author):

In this revised manuscript by Hollin et. al., authors used a newer proteome-wide approach, called R-DeeP, to characterize RNA dependent protein interactions in malaria parasite *Plasmodium falciparum*. Despite their central role in RNA function, the RNA-binding specificities of many RNA-dependent proteins remain unknown or incompletely defined in higher and lower eukaryotes. To address this, the authors have assembled a genome-scale collection of RDPs and RBPs and their RNA-binding domains (RBDs) and quantitatively assessed their specificities. This is a timely contribution from a research group with significant experience in malaria RNP analysis and will be a valuable contribution to the field. Validation of some of the findings could be a concern, however, genetic manipulations with *falciparum* is notoriously hard and time consuming and could take significant time to resolve. Overall, I'm satisfied with the revised version of the manuscript.

We thank the reviewer for his positive comment.

Reviewer #4 (Remarks to the Author):

The first question raised by the Reviewer 2 is indeed quite crucial since all the following quantitative analyses are strongly dependent on this first step. For the missing values, the authors effectively imputed the same values found in the other replicate. This seemingly simple calculation, however, represents many bold assumptions. Not being detected in one sample suggests low abundance and it is not clear how to correctly impute their abundance. When not established, I would suggest to take a rather conservative calculation, e.g., take the average. If taking average severely affects the whole analysis, it means that the whole finding is dependent on this bold imputation step. To clearly demonstrate the

power of R-Deep, the author need to take average or even analyze each replicate separately and take intersection or union with a proper statistical validation.

We have followed the reviewer's advice and modified our analysis to take the average of the two replicates. A total of 898 proteins were now significantly shifted, including 759 (96%) that were previously determined as significant. Only 26 were lost and 139 new proteins were now considered as statistically significant. We re-analyzed all GO enrichment, molecular weight changes, correlation and RNase-dependance of all complexes. For each step only minor changes were observed with no impact on our conclusions. The manuscript and figures were updated with the new numbers and statistics.

On the other hand, the data analysis for peptide and protein identification was done in a very clear and rigorous way so I doubt that the proteins found contain many false positive identifications. For identification, taking union proteins from different replicates would not bring any issue as the FDR control has been done jointly using a well-established method, which is Percolator. But quantification is another issue and imputation is one of the topics to which the field still struggles to find the correct answer.

We thank the reviewer for this assessment of our analysis, and we agree that, in general, imputation is a challenge for the field.

Related to this, the Reviewer 2 raised the second question on the reproducibility between replicates. Already the authors have the separate results from each replicate and could proceed with the follow up analysis. If the candidate PF3D7_0823200 was lost during this procedure, it suggests that this imperfect imputation was introducing the candidate. I agree that the candidate is a true RBP. But the authors could not argue strongly that their method was so powerful that this candidate has been found. I believe the method is one of the major points in this manuscript (while R-Deep already has been published) and should be robust for other applications. Even if the authors have to use a bit higher FDR (e.g., 3%) to find the valid candidates, I would recommend to do so since FDR is something that we know how to control while such a bold imputation is out of control.

As indicated above, we have eliminated the "bold imputation" approach from our previous submission, replacing it with a simple average. We hope that this addresses the reviewer's concern. We are reluctant to, in addition, analyze each replicate separately because we would then need to adjust our significance analysis for these three tests (the average plus the two replicates). With the average method, our candidate PF3D7_0823200 was still detected as significant and all candidates selected for antibody production and western blot analysis were conserved in their respective category (shifted or not shifted).

Regarding the question 4: it is very interesting to see that the RNAase sensitivity varies for different RBPs. To address this, the authors suggested a hypothesis "we can hypothesize that in our experimental conditions, the RNA may be protected from RNase activity and only ribosomal proteins located on the surface of the complex could have been detached from the rest during the RNase treatment." And then they reports a possible limitation, which I believe raise the overall quality of the manuscript. However, I would recommend to give a few selected raw signals that showing this new hypothesis is indeed the case. Since this is not only interesting and quite counter intuitive, I believe the authors should provide (at least) a few selected spectra that can help the readers to be convinced.

We are sorry but we are not sure we understand what the reviewer wants to see with the raw spectra. Instead, we provided NSAF values for several ribosomal proteins (Supplementary Fig 1b). If this figure does not address the reviewer comment, please clarify and we'll be happy to provide the information.

I believe other concerns raised by Reviewer 2 have been addressed.

We thank the reviewer for this comment.